# Active force generation contributes to the complexity of spontaneous activity and to the response to stretch of murine cardiomyocyte cultures

Seyma Nayir[1], Stéphanie P. Lacour[2] and Jan P. Kucera[1]

[1]*Department of Physiology, University of Bern, Bern, Switzerland*
[2]*Laboratory for Soft Bioelectronic Interfaces, EPFL, Geneva, Switzerland*

Edited by: Bjorn Knollmann & Michael Shattock

The peer review history is available in the Supporting Information section of this article (https://doi.org/10.1113/JP283083#support-information-section).

**Abstract**  Cardiomyocyte cultures exhibit spontaneous electrical and contractile activity, as in a natural cardiac pacemaker. In such preparations, beat rate variability exhibits features similar to those of heart rate variability *in vivo*. Mechanical deformations and forces feed back on the

**Seyma Nayir** is currently a PhD candidate in the Integrative Cardiac Bioelectricity Group led by Professor Jan P. Kucera at the Department of Physiology of the University of Bern. She has a background in Experimental Physics, with a Bachelor's and Master's degree from Istanbul Technical University (Turkey). Seyma really enjoys doing technically challenging experiments and she is keen to apply her knowledge to life sciences. Her research focuses on the effects of mechanical deformation (stretch) on cardiac electrical activity.

The Journal of Physiology

electrical properties of cardiomyocytes, but it is not fully elucidated how this mechano-electrical interplay affects beating variability in such preparations. Using stretchable microelectrode arrays, we assessed the effects of the myosin inhibitor blebbistatin and the non-selective stretch-activated channel blocker streptomycin on beating variability and on the response of neonatal or fetal murine ventricular cell cultures against deformation. Spontaneous electrical activity was recorded without stretch and upon predefined deformation protocols (5% uniaxial and 2% equibiaxial strain, applied repeatedly for 1 min every 3 min). Without stretch, spontaneous activity originated from the edge of the preparations, and its site of origin switched frequently in a complex manner across the cultures. Blebbistatin did not change mean beat rate, but it decreased the spatial complexity of spontaneous activity. In contrast, streptomycin did not exert any manifest effects. During the deformation protocols, beat rate increased transiently upon stretch but, paradoxically, also upon release. Blebbistatin attenuated the response to stretch, whereas this response was not affected by streptomycin. Therefore, our data support the notion that in a spontaneously firing network of cardiomyocytes, active force generation, rather than stretch-activated channels, is involved mechanistically in the complexity of the spatiotemporal patterns of spontaneous activity and in the stretch-induced acceleration of beating.

(Received 10 March 2022; accepted after revision 1 June 2022; first published online 9 June 2022)

**Corresponding author** Jan P. Kucera: Department of Physiology, University of Bern, Bühlplatz 5, CH-3012 Bern, Switzerland. Email: jan.kucera@unibe.ch

**Abstract figure legend** Mechano-electrical feedback modulates myocardial electrical function, including pacemaking. By growing monolayer cultures of spontaneously active murine cardiac cells on stretchable microelectrode arrays, we examined whether active contractions influence the spatiotemporal characteristics of beating variability and the effects of stretching on beat rate. In control conditions (no stretch and no pharmacological agent), the origin of the electrical activity changed frequently. After blocking contractions with blebbistatin, the spatiotemporal pattern of electrical activity became less variable and less complex. In control conditions (no pharmacological agent), stretching (and also releasing) the cardiomyocyte monolayers increased the beat rate transiently. Blebbistatin attenuated the acceleration of beating upon stretch. In contrast, streptomycin had no detectable effects. Thus, active force generation is involved in determining beating variability in spontaneously active cardiac tissue. Possible mechanisms might include cellular processes that sense contraction and chemical messengers. Our study contributes to the understanding of how mechano-electrical feedback influences heart rate variability.

## Key points

- Monolayer cultures of cardiac cells exhibit spontaneous electrical and contractile activity, as in a natural cardiac pacemaker. Beating variability in these preparations recapitulates the power-law behaviour of heart rate variability *in vivo*. However, the effects of mechano-electrical feedback on beating variability are not yet fully understood.

- Using stretchable microelectrode arrays, we examined the effects of the contraction uncoupler blebbistatin and the non-specific stretch-activated channel blocker streptomycin on beating variability and on stretch-induced changes of beat rate.

- Without stretch, blebbistatin decreased the spatial complexity of beating variability, whereas streptomycin had no effects.

- Both stretch and release increased beat rate transiently; blebbistatin attenuated the increase of beat rate upon stretch, whereas streptomycin had no effects.

- Active force generation contributes to the complexity of spatiotemporal patterns of beating variability and to the increase of beat rate upon mechanical deformation. Our study contributes to the understanding of how mechano-electrical feedback influences heart rate variability.

## Introduction

During normal heart rhythm, every heart beat is initiated by an action potential (AP) that originates in the sino-atrial node (SAN). The AP then propagates through the heart and triggers the contraction of cardiomyocytes (Bers, 2002). Irregular or uncoordinated electrical excitation can result in arrhythmias and compromise cardiac output.

The SAN is a network of connected cells that generate APs spontaneously by slow diastolic depolarization that brings the membrane potential to the threshold of voltage-gated inward currents (Noma, 1996). Two jointly operating mechanisms, known as the 'membrane clock' and the 'calcium clock', contribute to this diastolic depolarization (Lakatta & DiFrancesco, 2009). The membrane clock relies on the activation of the inward 'funny' current carried by hyperpolarization-activated nucleotide-gated cation channels HCN4 (Bychkov et al., 2020; DiFrancesco, 2010), and the $Ca^{2+}$ clock results from the spontaneous release of $Ca^{2+}$ from the sarcoplasmic reticulum during late diastole, causing additional inward current via sodium–calcium exchange (Maltsev & Lakatta, 2007). In the classical model of SAN function, pacemaker cells synchronize via gap junctional coupling by mutual entrainment, whereby the cells or a group of cells exhibiting the fastest intrinsic rate entrain the other cells of the tissue (Jalife, 1984; Verheijck et al., 1998).

However, the SAN is a very heterogeneous structure, with different patterns of ion channel expression and with an intercellular coupling gradient of different types of connexins (Boyett et al., 2000). Connexin 43 (Cx43) is expressed at the periphery of the SAN, but it is scarce in its centre, where connexin 40 (Cx40) mediates intercellular coupling (Boyett et al., 2000). Detailed structural studies have revealed that the SAN is formed by a meshwork of intermingling Cx43-positive and Cx43-negative strands (Boyett et al., 2000; Bychkov et al., 2020), whereby the Cx43-positive strands form pathways along which APs are conducted to the surrounding atrium in a discontinuous manner (Bychkov et al., 2020). This mingling of Cx43-positive and Cx43-negative tissue provides a mechanism by which the SAN is able to drive the surrounding atrium, which, by exhibiting a more negative resting membrane potential, tends to suppress pacemaking (Boyett et al., 2000; Bychkov et al., 2020; Joyner & van Capelle, 1986). Recent investigations demonstrated a meshwork of HCN4-positive, F-actin-negative and Cx43-negative cells intertwined with a meshwork of HCN4-negative, F-actin-positive and Cx43-positive cells (Bychkov et al., 2020). At the large interface between both meshworks, these two cellular populations are joined by a very narrow gap (Bychkov et al., 2020). This intermeshing extends over the entire SAN, without gradient or mosaic patterns, and the HCN4-positive cells exhibit complex shapes (Bychkov et al., 2020).

At the functional level, it was shown in the same study (Bychkov et al., 2020) that synchronized APs emerge from heterogeneous subcellular subthreshold $Ca^{2+}$ signals (local $Ca^{2+}$ releases) in the HCN4-positive and F-actin-negative cellular meshwork. These local $Ca^{2+}$ releases differ greatly in spatial distribution, frequency, amplitude and phase. At the cellular level, patterns of $Ca^{2+}$ signalling range from highly synchronized to less synchronized or irregular. Some SAN cells generate diastolic local $Ca^{2+}$ releases preceding full $Ca^{2+}$ transients in adjacent cells, but do not produce full $Ca^{2+}$ transients on their own. The resulting $Ca^{2+}$ signals thus appear discontinuous at a macroscopic scale, and the interaction of these $Ca^{2+}$ events leads to a multiscale process of impulse generation. The HCN4-positive cells are believed to be the region of weakly coupled entrained oscillators and, possibly, only a fraction of HCN4-positive cells participate in AP generation.

Using patch-clamp recordings in isolated SAN cells from wild-type mice and a knock-in murine model of HCN4 with defective regulation by cyclic adenosine monophosphate (cAMP), another research group revealed two populations of SAN cells: one population of cells firing APs and another one that switches between a firing and a non-firing mode (Fenske et al., 2020). Using $Ca^{2+}$ mapping, the authors correlated these populations with one exhibiting large global $Ca^{2+}$ transients, and the other with only subthreshold local $Ca^{2+}$ signals or $Ca^{2+}$ waves that did not spread to other cells. The proportion of non-firing cells was larger in the knock-in model, indicating that regulation by cAMP is important to bring the cells to fire, and the proportion of the two pools was modulated by agonists and antagonists of the autonomous nervous system. Based on these findings, the authors proposed that the non-firing cells form an inhibitory cell pool that exerts a tonic depressing effect on the firing cell pool, thus decreasing the rate of the firing cells. Thus, besides classical mutual entrainment (Jalife, 1984), SAN activity might also by modulated by this tonic interaction, leading to a new model of chronotropy, in which the autonomous nervous system influences the relative proportion of the firing and non-firing populations (Fenske et al., 2020).

At the level of functional integration, the SAN activity thus results from the self-similar organization of local $Ca^{2+}$ signals from the subcellular scale to the tissue scale, in the form of a process that can be envisioned as fractal-like (Bychkov et al., 2020). These findings challenge the view that a dominant centre in the SAN drives pacemaking (Clancy & Santana, 2020). Rather, APs emerge from very heterogeneous self-organizing $Ca^{2+}$ events arising in very diverse cell populations that operate

in a subthreshold regimen and interact via different coupling modalities, which permits rapid entrainment and stochastic resonance (Clancy & Santana, 2020). In the crucial regimen, the signals arising from the $Ca^{2+}$ clock eventually percolate through the SAN (Weiss & Qu, 2020), and it has been proposed that this mode of operation confers on the SAN a high level of resilience and tunability (Clancy & Santana, 2020). Taken together, these studies have provided important insight into the interdependence of the $Ca^{2+}$ and voltage clock mechanisms and the self-organized rhythmic activity of the SAN.

Heart rate is modulated mainly by the autonomic nervous system. The analysis of the resulting heart rate variability (HRV) is an established method to evaluate autonomous nervous system function and to derive prognostic markers (Camm & Malik, 1996). Interestingly, at very low frequencies, HRV exhibits a power-law behaviour (Akselrod et al., 1981; Camm & Malik, 1996; Sassi et al., 2015), which implies that HRV exhibits statistically self-similar (fractal) properties without any characteristic time scale. It was reported that the exponent of this power law and related fractal measures [such as the scaling exponent of detrended fluctuation analysis (DFA); Behar et al., 2018a; Peng et al., 1995] might be of diagnostic or predictive value in the context of cardiac disease (Gang et al., 2011; Makikallio et al., 2001a, 2001b). Remarkably, this fractal behaviour is present even in the isolated and denervated heart and in the isolated SAN (Yaniv et al., 2014). Previously, we have demonstrated a power-law behaviour of beat rate variability (BRV) in monolayer cultures of neonatal rat ventricular myocytes (Kucera et al., 2000; Ponard et al., 2007), an *in vitro* model of a natural cardiac pacemaker, and such fractal behaviour of BRV was also reported in isolated cardiomyocytes (Harada et al., 2009) and in embryoid bodies of human embryonic and induced pluripotent stem cell-derived cardiomyocytes (Mandel et al., 2012). This fractal behaviour of HRV and BRV thus appears to be a universal phenomenon that is mediated, at least in part, by intrinsic dynamic mechanisms at the tissue and cell levels. It might be caused by fluctuations in ion currents owing to their stochastic gating (Wilders & Jongsma, 1993), by stochastic local $Ca^{2+}$ releases from the sarcoplasmic reticulum (Monfredi et al., 2013) or by spatial and temporal variations of ion channel expression and turnover (Ponard et al., 2007). Possibly, criticality and self-organization of single intracellular calcium release events into larger waves contribute to the power-law behaviour of BRV by modulating the calcium clock (Nivala et al., 2012).

In parallel to the continuous variation of beating rate, cardiac activity is affected continuously by mechanical influences resulting from passive forces and active contractions. The typical example is the Frank–Starling mechanism, which permits, on a beat-to-beat basis, the stroke volume to be adjusted and optimized to diastolic filling (Lookin & Protsenko, 2019). Regarding pacemaker function, it is known that an increase in right atrial pressure causes an increase of heart rate. Although the regulation of heart rate, which also contributes to adjusting cardiac output to variations in venous return, involves the autonomous nervous system (Bainbridge reflex), it is also, in part, intrinsic to the SAN and mediated by stretch-activated channels (SACs; Cooper & Kohl, 2005; Kohl et al., 1994).

The modulation of electrophysiological characteristics by mechanical influences is generally known as mechano-electrical feedback (MEF; Kamkin et al., 2005; Kohl et al., 1994, 2006; McNary et al., 2008; Quinn et al., 2014; Riemer & Tung, 2003). Mechano-electrical feedback is most frequently attributed to non-selective SACs (Huang et al., 2009; McNary et al., 2008), which cause depolarization and triggered activity when activated during diastole (Quinn et al., 2017; Riemer & Tung, 2003). Stretch-activated channels can be present constitutively in cardiomyocytes or in fibroblasts electrically coupled to myocytes (Grand et al., 2014; Kohl et al., 1994; Quinn et al., 2014). Myocardial stretch also modulates other ion channels directly (Beyder et al., 2010; McNary & Sachse, 2009) or indirectly by involving the cytoskeleton and intracellular signalling mediators, such as nitric oxide NO (Boycott et al., 2020). In addition, MEF can also be mediated by changes in passive tissue resistance (McNary et al., 2008; Sachse et al., 2004) or capacitance (Mills et al., 2008; Pfeiffer et al., 2014). Given that MEF can cause arrhythmias, it is important, in both physiology and cardiology, to understand the relationship between mechanical effects and the excitation of cardiac tissue.

Previous studies focused mainly on passive mechanical influences, whereas the direct involvement of active force generation in MEF has, in comparison, scarcely been investigated. In this context, it was observed in cardiomyocyte cultures that the myosin inhibitor and contraction uncoupler blebbistatin reverts the pro-arrhythmic slowing of conduction induced by co-cultured myofibroblasts (Thompson et al., 2011), thereby suggesting that intercellular mechanical junctions (e.g. adherens junctions) between these cells might transmit active forces to mechanosensitive channels (Thompson et al., 2014). However, this reversal of conduction slowing was not corroborated in similar experiments by others (Grand et al., 2014), suggesting that SACs in the myofibroblasts primarily contribute to stretch-induced conduction slowing. In whole hearts, it was observed that blebbistatin decreases the complexity of ventricular fibrillation (Brines et al., 2012). In the setting of pacemaker activity, it was reported that cultured cardiomyocyte monolayers exhibit a higher spatial and temporal instability of spontaneous activation rate when grown on elastic substrates compared with rigid glass

(Boudreau-Béland et al., 2015). It was also shown that spontaneously beating cardiomyocytes cultured on an elastic substrate can synchronize even without direct contact (no gap junctional coupling) if they are sufficiently close together, and this synchronization is disrupted when active force generation is blocked (Nitsan et al., 2016; Viner et al., 2019). In the context of the SAN, it is debated presently whether the HCN4-positive cells communicate exclusively via gap junctions or whether other factors, such as mechanical signalling, chemical signalling (e.g. NO) or even ephaptic coupling (Ly & Weinberg, 2022), are involved (Bychkov et al., 2020; Weiss & Qu, 2020).

In the present work, to gain a better understanding of the role of MEF in modulating pacemaker activity, we examined, in a first step, how the spatiotemporal patterns of spontaneous electrical activity of murine ventricular cell cultures grown on stretchable micro-electrode arrays (sMEAs) are influenced by blebbistatin and the SAC blocker streptomycin. Although these preparations do not mimic the extreme structural and functional complexity of the SAN, they nevertheless recapitulate important hallmarks of pacemaker function, in particular the combination of voltage and calcium clocks (Ponard et al., 2007) and the power-law behaviour of BRV, which is typical for self-critical and self-organized processes (Kucera et al., 2000; Ponard et al., 2007; Weiss & Qu, 2020). In a second step, we investigated how the response of beat rate to acute stretch is modulated by these two pharmacological agents. Beat rate was increased transiently by stretch but, interestingly, also by its release. Blebbistatin decreased the complexity of beating variability and attenuated the increase of beat rate upon stretch. Surprisingly, streptomycin did not exert any detectable effects. Our study thus unveils new roles of active force generation in beating variability and MEF.

## Methods

### Ethical approval

Animals were handled in accordance with the ethical principles and guidelines of the Swiss Academy of Medical Sciences. The procurement of animals, the husbandry and the experiments conformed to the European Convention for the Protection of Vertebrate Animals used for Experimental and other Scientific Purposes. The protocols were reviewed and authorized by the Commission of Animal Experimentation of the Cantonal Veterinary Office of the Canton of Bern, Switzerland (authorization no. BE36/19), in accordance with Swiss legislation. Experiments were carried out according to the guidelines laid down by the animal welfare committee of the University of Bern and conformed to the principles and regulations described by Grundy (2015).

### Stretchable microelectrode arrays

Polydimethylsiloxane (PDMS)-based sMEAs were micro-fabricated using a previously published process (Buccarello et al., 2018). In brief, 100-mm silicon wafers were spin-coated with poly(4-styrenesulfonic acid) (PSS; 18 wt. % in $H_2O$; Sigma-Aldrich). Subsequently, a 0.5-mm-thick PDMS base layer was spun onto the wafers. The PDMS (Sylgard 184; Dow Corning) was prepared at a mass ratio of 10:1 of base to curing agent and cured for $\geq 2$ h at 75–80°C. To prepare electrodes, interconnectors and markers for the monitoring of strain (Fig. 1A), a first 5-nm-thick chromium layer and a second 35-nm-thick gold layer were thermally evaporated (Auto 306; Edwards) onto the PDMS base layer through a laser-machined poly-imide mask. Approximately 10-$\mu$m-thick encapsulation layers were prepared separately by spinning (5000 r.p.m. for 1 min) PDMS on another 1- to 2-mm-thick PDMS carrier, which was spin-coated beforehand with diluted PSS (9% wt. % in $H_2O$) and dried at room temperature. After curing, the encapsulation layers were perforated using punchers at the contact sites of the electrodes and cut into 2 cm $\times$ 2 cm pieces. The encapsulation layers were then aligned and bonded onto the sMEAs after activation using oxygen plasma. Finally, the wafers carrying the sMEAs were immersed for 2–4 h in distilled water, whereby the dissolution of PSS led to the spontaneous detachment of the sMEAs from the wafers, and of the PDMS carrier layers supporting the encapsulation layers. After drying, the sMEAs were cut into their final shape (see Fig. 1A). The electrode layout consisted of four stimulation dipoles (diameter, 0.95 mm) and 12 unipolar recording electrodes (diameter, 0.2 mm) separated by 2 mm (Fig. 1A).

### Murine cardiac cell cultures on sMEAs

Cultures of ventricular myocytes from neonatal (0–1 day postpartum) or fetal (19.5 days postcoitum) wild-type C57BL/6J mice (Charles River) were prepared according to previously published protocols (Beauchamp et al., 2004; Buccarello et al., 2018; Prudat & Kucera, 2014). The animals had *ad libitum* access to food and water. Neonates were killed by decapitation. To collect fetuses, the mother animal was terminally anaesthetized by intra-peritoneal injection of xylazine and ketamine (10 and 80 mg/kg body mass, respectively). After disappearance of the pedal reflex, the animal was exsanguinated and the uterus extracted. The fetuses were then collected and killed by decapitation. A total of 56 neonates, 52 fetuses and 11 adult animals were used. The hearts were extracted from the donors, and the ventricles were minced to 1 mm pieces. Cardiomyocytes were then dissociated enzymatically using trypsin (0.075%; Gibco) and pancreatin (100 mg/l; Sigma-Aldrich) in Hanks'

balanced salts solution (HBSS; Bioconcept) containing Phenol Red at 37°C. All cells were pooled, irrespective of the sex of the animals. Subsequently, the cell suspension was centrifuged, resuspended in enzyme-free culture medium and preplated for 2 h to minimize the myofibroblast content. Thereafter, the cells were counted using a Neubauer chamber (Bioswisstech).

In parallel, the sMEAs were preconditioned by oxygen plasma activation and coating with type I collagen (Sigma-Aldrich), diluted at a concentration of 200 mg/l in HBSS (Gibco) containing 12% acetic acid (Grogg Chemie). To generate disc-shaped monolayers, polyimide masks with a circular hole 8 mm in diameter were centred and positioned on the sMEA electrode layouts (see Fig. 1A), and the collagen solution was applied over the masks. To build culture chambers, hollow PDMS cylinders (inner diameter, 17.5 mm; outer diameter, 22 mm; and height, 15 mm) were centred and affixed onto the sMEAs using Vaseline. During stretch experiments, this permitted the cylinders to slide on the sMEAs without interfering with their deformation.

After washing the culture substrates with HBSS and sterilizing them using ultraviolet light, the cardiomyocytes were seeded at a density of $1.7 \times 10^5/cm^2$ and incubated at 36°C in air with 0.9% $CO_2$. The culture medium (M199 with Hanks' salts; Sigma-Aldrich) was supplemented with 10% neonatal calf serum (Amimed), 10 mmol/l Hepes (Gibco) and 0.68 mmol/l L-glutamine (Sigma-Aldrich). To prevent bacterial proliferation, the medium was also supplemented with 20 mg/l streptomycin and 20 000 U/l penicillin (Biochrom). Of note, the effect of streptomycin as an SAC blocker is reversible upon washout (Ninio & Saint, 2008). To minimize myofibroblast proliferation, 100 $\mu$mol/l bromodeoxyuridine (Sigma-Aldrich) was also added to the medium. After 24 h, the polyimide masks were removed, and the culture medium was changed (with 5% instead of 10% neonatal calf serum) to remove non-attached cells. A microphotograph of a culture is shown in Fig. 1A.

## Stretching system

The stretching system, described in detail previously (Buccarello et al., 2018), consisted of four linear motorized stages (MTS25-Z8; Thorlabs) arranged in a symmetric manner along four perpendicular directions. The stages carried supports for printed circuit boards. These printed circuit boards were attached on the stages using screws (passed through holes punched in the sMEAs, as visible in the top photograph of Fig. 1A) and served to clamp the edges of the sMEAs and to interface their connecting pads with a custom-made amplifier array (Kondratyev et al., 2007).

Pairwise concurrent operation of the stages permitted the sMEAs to be stretched independently along their two main axes [horizontal ($x$) and vertical ($y$) directions in Fig. 1A]. The markers integrated in the sMEAs were imaged by a camera (B910 HD Webcam; Logitech) positioned underneath. A custom MATLAB (MathWorks) analysis program identified the markers in real time, which enabled us to monitor the applied strain by computing the average deformation gradient tensor in the centre of the sMEA and to adjust the strain to the desired target strain using a closed control loop (Buccarello et al., 2018). In particular, the system made it possible to apply strictly uniaxial strains in the $x$ and $y$ directions by precisely compensating the tendency to constriction of an elastic material along axes orthogonal to the main direction of stretch.

## Electrophysiological experiments

Experiments were conducted 2–3 days after seeding the cultures. To ensure identical conditions in all experiments, the culture medium was first replaced with HBSS (Gibco) prewarmed to 36°C. The sMEA was then mounted on the stretching system, and a ring-shaped gold wire was inserted into the bath to serve as an earth electrode. The system was then encased in a thermally insulating polystyrene box, in which a temperature of 37°C was maintained with humidified air using a controller (The Cube; Life Imaging Services). A period of 30 min was allowed for equilibration. Unipolar extracellular electro-grams from the recording electrodes (Fig. 1B, C and D) were amplified (gain, ×1000) and digitized at a sampling rate of 10 kHz and with 12-bit resolution using a data-acquisition processor (DAP 4400a; Microstar Laboratories).

For every preparation, a 15–30 min recording of spontaneous electrical activity was first conducted without any mechanical deformation. Subsequently, a second 15 min recording was conducted, during which, every 3 min, 5% uniaxial strain in the $x$ direction was applied and maintained for 1 min, followed by release to the undeformed configuration for 2 min; this protocol was repeated four or five times during the recording. Thereafter, a third and a fourth 15 min recording were conducted using the same protocol, but respectively with 5% uniaxial strain in the $y$ direction and 2% biaxial strain (i.e. in both directions simultaneously).

Owing to the finite speed of the linear stages, completing the full stretch and releasing back to the initial undeformed state took ∼2 s (uniaxial protocols) and 1.5 s (biaxial protocols). To have precise time stamps of the applied stretches and subsequent releases, electric pulses were generated by the stretch control program at the onset and at the end of every operation of the linear motor stages, and this pulse was fed into an additional data-acquisition channel via the audio socket of the computer.

In additional series of experiments, the same entire procedure was repeated after the addition of blebbistatin (10 $\mu$mol/l; Sigma-Aldrich) or streptomycin (20 $\mu$mol/l; Calbiochem) to the culture bath. Blebbistatin is a myosin II-specific inhibitor that blocks the cross-bridge cycle by binding to an allosteric site in the motor domain of myosin II. It stabilizes an actin-detached state of myosin II, which prevents it from generating force and using ATP (Rauscher et al., 2018; Roman et al., 2018). We selected blebbistatin because this agent is widely used by the cardiac optical mapping community to suppress motion artefacts (e.g. Fenske et al., 2020; Lou et al., 2012; Quinn et al., 2017) and exerts less effect on electrophysiological parameters compared with other agents, such as 2,3-butanedione monoxime, diacetyl monoxime or cytochalasin D, which are known to modify the cardiac action potential (Baker et al., 2004; Brines et al., 2012; Lou et al., 2012). Streptomycin is an aminoglycoside antibiotic that also blocks non-specific SACs (Quinn & Kohl, 2021). We selected streptomycin because it is a common pharmacological tool to investigate SACs in cardiac electrophysiology (Quinn et al., 2017; Quinn & Kohl, 2021; White, 2006) and it has been used successfully in cardiac cell culture systems similar to ours (Grand et al., 2014; Thompson et al., 2011). Other widely used agents are gadolinium and GsMTx-4. However, gadolinium precipitates in the presence of phosphate and bicarbonate (Quinn & Kohl, 2021), precluding its use in the medium that we used for experiments (HBSS). It might also block the sodium current (Quinn & Kohl, 2021).

### Data analysis

All analyses were conducted using MATLAB. For every recording channel, the signal was first median filtered (sliding window of three samples) to remove occasional single-sample glitches, then filtered using a digital AC coupling filter to remove baseline drift (time constant, 3 ms) and, if the signal-to-noise ratio was low, low-pass filtered using a convolutional Kaiser filter (cut-off frequency 0.5–3 kHz). Activation times were identified as the time points of the minimum of the first derivative, which was computed using a centred finite-difference scheme. All recordings were inspected and curated manually using a custom MATLAB graphical user interface to correct false-negative detections and remove false positives. The user interface permitted us to adjust thresholds for detection, whereby the lowest possible threshold was always used to ensure a high sensitivity (low number of false-negative detections that needed to be corrected manually) at the expense of a lower specificity (larger number of detections that needed to be deleted manually).

Signals had a typical biphasic shape (see examples in Fig. 1B–D), with an initial positive phase, a rapid down-stroke typically lasting <1 ms and a negative phase. Given that this time course occurs on a time scale shorter than the time constant of the AC filter and because the Kaiser filter involved a symmetric kernel (producing, by design, no phase shifts in the frequency domain), the resulting effect on activation time detection was minimal. Filtering the signal using the Kaiser filter improved the signal-to-noise ratio and thus the proportion of false-positive or false-negative detections that had to be curated manually. Example raw signals, the determination of activation time and the beneficial effects of filtering are illustrated in Fig. 1B.

In some traces, the extracellular electrograms were fractionated, i.e. they exhibited more than one local minimum of their first derivative. In this case, as illustrated in Fig. 1B, corresponding activation times occurring at an interval smaller than a preset value (1 ms) were fused by computing their weighted average, whereby the values of the derivative minima were used as weights.

Spontaneous electrical activity manifested as clusters of signals (Fig. 1C) with longer intervals between them, reflecting the rapid spread of excitation across the preparations without activity in between. This permitted us to identify individual spontaneous excitations unambiguously. To construct interbeat interval (IBI) time series, the mean activation time series ($m_i$) was first calculated for every excitation by averaging the activation times at the available electrodes as follows:

$$m_i = \frac{1}{N} \sum_{j=1}^{N} t_{i,j},$$

where $N$ is the number of electrodes, and $t_{i,j}$ is the activation time at electrode $j$ for the $i^{\text{th}}$ excitation. The IBI time series was then computed as the difference series of $m_i$.

In the spatial domain, activations typically originated at the periphery of the preparations (see Fig. 1D). Owing to the limited number of electrodes, it was not possible to locate the activation focus exactly. Therefore, we resorted to another quantitative marker, which we call slowness, to describe the excitation pattern globally. Slowness is the reciprocal of conduction velocity (de Lange & Kucera, 2009) and is expressed in seconds per centimetre. In two dimensions (or more), slowness can be understood as the spatial gradient of activation time, a vector with the same direction as conduction velocity, but a reciprocal magnitude. For every excitation, $i$, the average slowness, $\overrightarrow{s_i}$, was computed by least-squares fitting a two-dimensional linear function to activation time as follows:

$$t_{i,j} = t_{i,0} + s_{i,x} x_j + s_{i,y} y_j,$$

where $t_{i,0}$, $s_{i,x}$, and $s_{i,y}$ are the fit parameters, and $x_j$ and $y_j$ are the coordinates of the $j^{\text{th}}$ electrode. Slowness was then

obtained as follows:

$$\vec{s}_i = \begin{pmatrix} s_{i,x} \\ s_{i,y} \end{pmatrix}.$$

Slowness, $\vec{s}$, is related to velocity $\vec{v}$ as $\vec{v} = (1/\|\vec{s}\|^2)\vec{s}$ (Bayly et al., 1998; Masè et al., 2021; van Schie et al., 2021). Our approach is essentially similar to that proposed by Bayly et al. (1998). Of note, slowness is the primary result of the fitting procedure, whereas velocity is always computed from slowness in a subsequent step (Bayly et al., 1998; Masè et al., 2021; van Schie et al., 2021). A second argument in favour of slowness is that if differences in activation times are small or if the excitation pattern is symmetric (e.g. a radially symmetric pattern originating in the centre of the mapped region), overall slowness will be very small (or even zero), and velocity will diverge towards infinity (division by zero; Bayly et al., 1998). The use of slowness permits us to avoid this singularity. Figure 1*D* illustrates activation maps and corresponding slowness vectors.

As an alternative, we used principal components analysis (PCA) to characterize the activation patterns. Initially, the activation times were offset by mean activation time, $m_i$, as follows:

$$t'_{i,j} = t_{i,j} - m_i.$$

The PCA was then conducted on $t'_{i,j}$. Unless specified otherwise, results are presented in terms of slowness.

The irregularity (complexity) of slowness during a recording of spontaneous activity was quantified using Shannon's entropy (Shannon, 1948) of the distribution of the tip of the slowness vector in the $x$–$y$ plane. The latter was partitioned into 0.005 s/cm × 0.005 s/cm square bins, and the fraction of data falling into a bin was determined as $P = n/\Sigma n$, where $n$ is the count in the given bin and $\Sigma n$ is the total count over all bins. Entropy ($E$) was then calculated as follows:

$$E = -\sum_{i,\, p_i \neq 0} p_i \ln p_i,$$

where $i$ represents the indices of the bins. Distributions that are spread out have a high $E$, whereas clustered or localized distributions have a lower $E$. We used $E$ as a quantitative marker of complexity. Although there exist distinct mathematical methods to quantify complexity (e.g. Lempel & Ziv, 1976), we use the words 'complex' and 'complexity' in their literal and intuitive meaning as 'complicated, not easy to describe or to understand', as opposed to 'simple'.

In the time domain, we used DFA to assess scale-free self-similar (i.e. fractal) features (Behar et al., 2018b; Peng et al., 1995). In brief, the series of IBIs were partitioned into segments of length $n$. In every segment, the IBIs were initially integrated (cumulatively summed), and the resulting series were detrended by least-squares fitting with a linear function. Then, the residual detrended fluctuation DF($n$) was computed as the average of the root mean square of all the detrended segments. The procedure was repeated for $n$ going from 4 to 128 in multiplicative steps by $\sqrt{2}$ ($n$ was rounded to the next integer), and the detrended fluctuation exponent $\alpha$ was computed as the linear regression slope of log[DF($n$)] *versus* log($n$). For white noise and Brownian noise, $\alpha$ is 0.5 and 1.5, respectively, whereas for $1/f$ fractal noise and HRV *in vivo*, $\alpha$ is near one (Behar et al., 2018b; Peng et al., 1995). The DFA was computed using a modified version of the corresponding MATLAB function from the PhysioZoo analysis package (Behar et al., 2018b).

Owing to BRV, discrete IBI time series are not sampled at constant time intervals. In analyses in which the response of beat rate to a given intervention (stretch or release) was averaged over repeated interventions, uniformly sampled instantaneous beat rate (IBR) series were constructed as follows. Initially, a function of time, $f(t)$, was defined using the series of mean activation times $m_i$ as follows:

$$f(t) = \frac{1}{m_{i+1} - m_i} \text{ for } m_i \leq t < m_{i+1}.$$

This function is constant between consecutive values of $m_i$ and discontinuous at the values of $m_i$. The IBR series $r_i$ uniformly resampled at constant intervals $\Delta t$ was then computed as follows:

$$r_i = \frac{1}{\Delta t} \int_{(i-1)\Delta t}^{i\Delta t} f(t)\, dt.$$

This resampling method offers the advantage that over any time interval, the sum of the corresponding values of $r_i$, multiplied by $\Delta t$, always corresponds (up to rounding to the next integer) to the number of IBIs, thus permitting averaging. We used a resampling frequency of 2 Hz ($\Delta t = 0.5$ s).

## Statistics

Summary data are presented as the mean ± SD or the median with interquartile range (IQR). Normality of distributions was assessed using the Shapiro–Wilk test. Normally distributed data were compared using Student's $t$ test (paired or one-sample, as appropriate). Non-normally distributed data were compared using the Wilcoxon signed rank test. All tests were two tailed. The threshold for significance was 0.05.

In correlation analyses, the ranks (i.e. quantiles) of the data were correlated instead of the data themselves to obtain Spearman's rank correlation coefficient, $\rho$.

All raw recordings and the MATLAB source code of the user interface and scripts for generating the panels of Figs 2–8 were deposited on a repository and are openly available (see Data Availability Statement).

## Results

### Spatiotemporal characterization of spontaneous activity in control conditions

In the first step, we characterized the spatiotemporal patterns of spontaneous electrical activity in disc-shaped monolayer cultures of murine ventricular myocytes grown on sMEAs without any mechanical deformation and pharmacological agent, in control conditions.

Figure 1*A* shows the design of the sMEAs and illustrates a cardiomyocyte culture. Figure 1*B* illustrates how activation times were identified. Figure 1*C* shows an example recording from the 12 unipolar electrodes of the sMEA. The electrograms illustrate the intrinsic variation of the IBI. For every electrical excitation (i.e. for every cluster of extracellular signals), we described the activation pattern by linear fitting of the measured activation times *versus* electrode positions. This fitting procedure yielded an overall slowness vector (which has the same direction as the overall conduction velocity vector, but a reciprocal magnitude). Figure 1*D* shows electrograms on an expanded time scale and corresponding activation maps of two consecutive activations, together with the slowness vector. Consistent with the findings of a previous study (Ponard et al., 2007), spontaneous activity originated from the edge of the preparations, and the site of origin switched frequently. This change in originating focus is illustrated in Fig. 1*D*.

The detailed analysis of spontaneous activity in an illustrative experiment is presented in Fig. 2. The IBI time series during the 15 min recording is shown in Fig. 2*A*. The IBI fluctuated irregularly from beat to beat over a wide range, exhibiting occasional sequences with short IBIs, in addition to longer pauses (one long pause occurred around 145 s with an isolated IBI of 6.07 s; the range of the plot in Fig. 2*A* was limited to 0–3 s to illustrate the IBI fluctuations better). Figure 2*B* shows the *x* and *y* components and the norm of the slowness vector for every excitation during the recording, and Fig. 2*C* shows the tip of the slowness vector and its trajectory in the *x*–*y* plane. These plots show that the slowness vector exhibited abrupt and irregular changes, indicating that during the recording the origin of the spontaneous electrical excitation changed frequently, in a complex manner. As an alternative method to investigate the change in activation pattern, we applied PCA of the activation times

of every excitation; Fig. 2*D* shows a two-dimensional plot of the first two principal components. In this PCA plot, excitation patterns that are highly correlated are clustered together. Thus, the PCA plot reveals the tendency of the activation patterns to change during the recording. Importantly, the pattern of the PCA plot and of slowness in the *x*–*y* plane are very similar, indicating that the slowness vector was sufficient to describe the activation patterns. This is supported by Fig. 2*E*, which shows that >95% of the variance was accounted for by the first two principal components, which is consistent with the two-dimensionality of the cell cultures. In all experiments together, the first two principal components always accounted for >90% of variance. Thus, in the following, we used slowness vectors to describe the activation patterns.

To assess the statistical self-similarity of the IBI series, we used DFA to obtain the self-similarity exponent $\alpha$ as shown in Fig. 2*F*. Figure 2*F* represents the root mean square of the detrended fluctuation (DF) *versus* segment length *n* in a double logarithmic plot. In this experiment, the slope $\alpha$ of the linear relationship between $\log[\text{DF}(n)]$ and $\log(n)$ was 0.76. This value being >0.5 indicates that the IBI variations exhibit statistically self-similar (fractal) properties distinct from random noise, similar to HRV *in vivo*, for which values near one were reported (Behar et al., 2018b; Peng et al., 1995). The fact that $\alpha < 1$ suggests that *in vivo*, further factors are likely to be involved in shaping HRV.

Figure 2*G* shows a plot of interbeat slowness differences (computed as the norm of the difference between successive slowness vectors) *versus* the IBIs themselves, and corresponding quantiles (represented as blue lines). The distribution was not normal and exhibited a cluster with small interbeat changes in slowness, and another cluster with large changes. This plot also shows that large interbeat slowness differences accounted for ∼20% of the data. To examine the relationship between interbeat slowness change and IBI further, we represented the same data in Fig. 2*H* in the form of a quantile–quantile plot. The greyscale map in the background shows the fraction of data within 0.2 × 0.2 quantile bins normalized by the fraction that would be expected in the absence of any correlation (0.2 × 0.2 = 0.04). Values above one, indicating a positive correlation, were striking in the top right and the bottom left corners of Fig. 2*H*. Spearman's rank correlation coefficient, $\rho$, was 0.169. The higher density of data points in these two corners and Spearman's $\rho$ being greater than zero thus reveal that interbeat slowness difference and IBI were positively correlated. Specifically, larger IBIs were associated with larger interbeat slowness differences, i.e. large changes in the origin of the spontaneous excitation, whereas smaller IBIs were associated with smaller changes in the activation pattern.

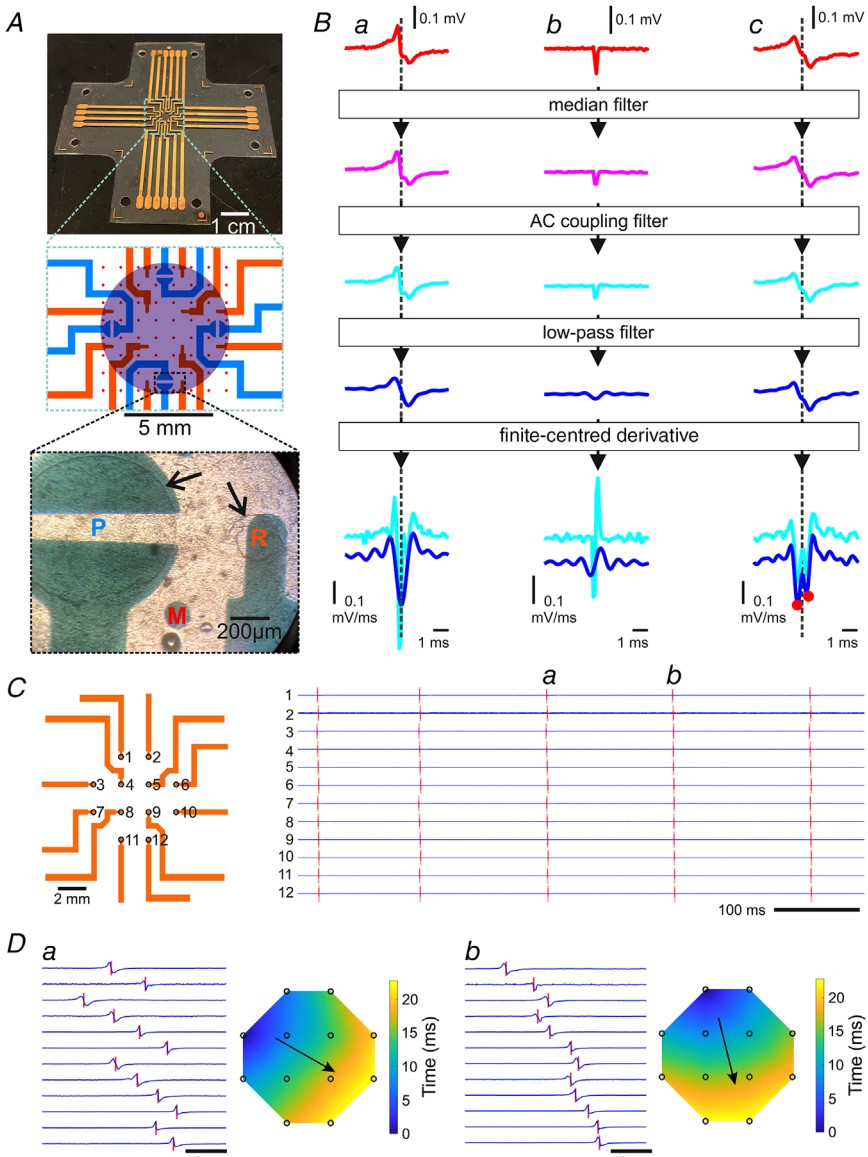

**Figure 1. Murine ventricular cardiomyocyte cultures on stretchable microelectrode arrays and example electrograms**

*A*, photograph of a stretchable microelectrode array (sMEA; top), with its schematic layout (middle). Colour code: blue, pacing dipoles; orange, recording electrodes; and red, markers for strain monitoring. The area where cells are seeded is shown as a purple disc. The bottom panel shows a photograph of a culture on an sMEA. Abbreviations: M, marker; P, pacing dipole; R, recording electrode. The borders of the encapsulation layer are visible as circles (black arrows). *B*, illustration of signal processing for a typical electrogram (*Ba*), an artefact resulting from an external perturbation (*Bb*) and a fractionated electrogram with two phases in its downstroke (*Bc*). Colour code: red, raw signals; magenta, after median filtering; cyan, after AC coupling; and blue, after low-pass filtering. The bottom panel shows derivatives of the signals before (cyan) and after (blue) low-pass filtering. Activation time is defined at the minimum of the derivative (dotted line in *Ba*). Differentiation of the signal in *Bc* resulted in two negative peaks of the derivative separated by <1 ms, and activation time was defined as the weighted average of their corresponding timings (red dots), whereby the amplitudes of the derivative peaks were used as weights. The artefact in (*Bb*) was identified as a false positive based on its clearly different time course. *C*, extracellular electrograms recorded by the set of 12 electrodes (schematic diagram and numbering on the left). Activation times are marked with red bars. *D*, electrograms and corresponding activation maps of two consecutive activations (*Da* and *Db*, as also labelled in *C*). The black arrows show the main direction of propagation, as calculated from the overall slowness vector (0.0341 and 0.0334 s/cm for *Da* and *Db*, corresponding to average conduction velocities of 29.3 and 29.9 cm/s, respectively). In *C* and *D*, the signals were normalized to their amplitude.

### Blebbistatin decreases the spatial complexity of spontaneous electrical activity and breaks down the correlation between IBIs and interbeat slowness differences

To evaluate the involvement of active cardiomyocyte contraction on the spatiotemporal characteristics of the variability of spontaneous electrical activity, we examined the effects of the myosin inhibitor blebbistatin. In these experiments, the analysis in Figure 2 was conducted for 15 min recordings (without external deformation) before and after the addition of 10 $\mu$mol/l blebbistatin to the preparations. Figure 3*A* shows the analysis of a control

recording (different monolayer culture from that in Fig. 2), and Fig. 3*B* shows the analysis of the recording with blebbistatin in the same preparation. Of note, we continue to use the term IBI, although blebbistatin blocked the contraction of the cardiomyocytes fully, as verified by microscope inspection. Comparison of the IBI series in control conditions and with blebbistatin revealed that larger excursions of IBI disappeared with blebbistatin.

In the spatial domain, abrupt changes of slowness were suppressed by blebbistatin, indicating that the origin of the electrical activation exhibited smaller and less frequent changes compared with control conditions. This was reflected by the more localized distribution of the

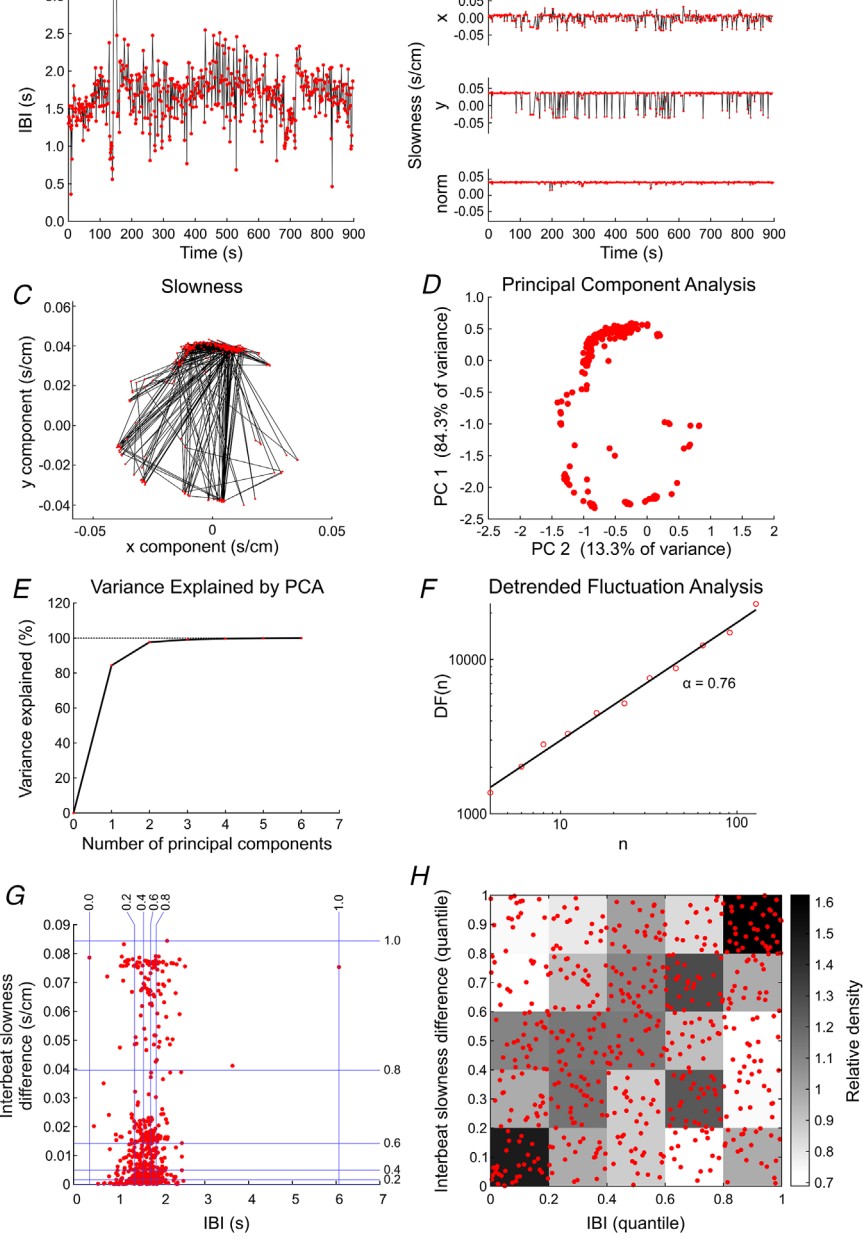

**Figure 2. Analysis of beating variability in an example preparation without pharmacological intervention or mechanical deformation**
*A*, interbeat interval (IBI) time series for a 15 min recording. *B*, *x* and *y* components and norm of the slowness vector for every activation. *C*, tip of the slowness vector (red dots) in the *x*–*y* plane. Black lines connect successive vector tips. *D*, principal components analysis (PCA) of the activation times of every excitation. *E*, cumulated variance explained by PCA as a function of the number of principal components. *F*, detrended fluctuation analysis (DFA) plot for the IBI time series. Abbreviations: $\alpha$, regression slope of DF(*n*) *versus n* on a double logarithmic scale; DF(*n*), detrended fluctuation for segments of length *n*; and *n*, length of the detrending segments. *G*, plot of interbeat slowness differences *versus* IBIs. The blue lines denote the 0, 0.2, 0.4, 0.6, 0.8 and 1.0 quantiles. *H*, Spearman correlation analysis between interbeat slowness difference and IBI (same data as in *G*, but plotted in terms of quantiles). The relative density was computed for every rectangle as the ratio of the effective number of data points to the number that would be expected without correlation. The Spearman's rank correlation coefficient, $\rho$, was 0.169.

slowness vector in the *x–y* plane and by the decrease of the entropy, *E*, of this distribution from 4.82 to 0.85. The DFA exponent, *α*, was 0.96 and 0.85 in control conditions and with blebbistatin, respectively.

The summary analysis of 11 experiments is presented in Fig. 3*C*. The corresponding recordings involved 547 ± 246 activations. Blebbistatin did not manifestly affect the mean IBI and thus the mean beat rate of the preparations ($P = 0.520$, Wilcoxon signed rank test). However, it significantly reduced the entropy of the distribution of the slowness vector ($P = 0.0293$, Student's paired *t* test),

and thus it decreased the complexity of the variability of the activation pattern. Blebbistatin did not have any effect on the exponent *α* ($P = 0.315$, Student's paired *t* test), indicating that the self-similarity of the IBI series was preserved. In control conditions, Spearman's *ρ* was positive in all experiments (different from zero with $P = 0.000112$, Student's one-sample *t* test *vs.* 0), confirming the correlation between the interbeat changes in activation patterns and the IBIs. Interestingly, in the presence of blebbistatin, the values of *ρ* clustered near zero ($P = 0.320$, Student's one-sample *t* test for a difference

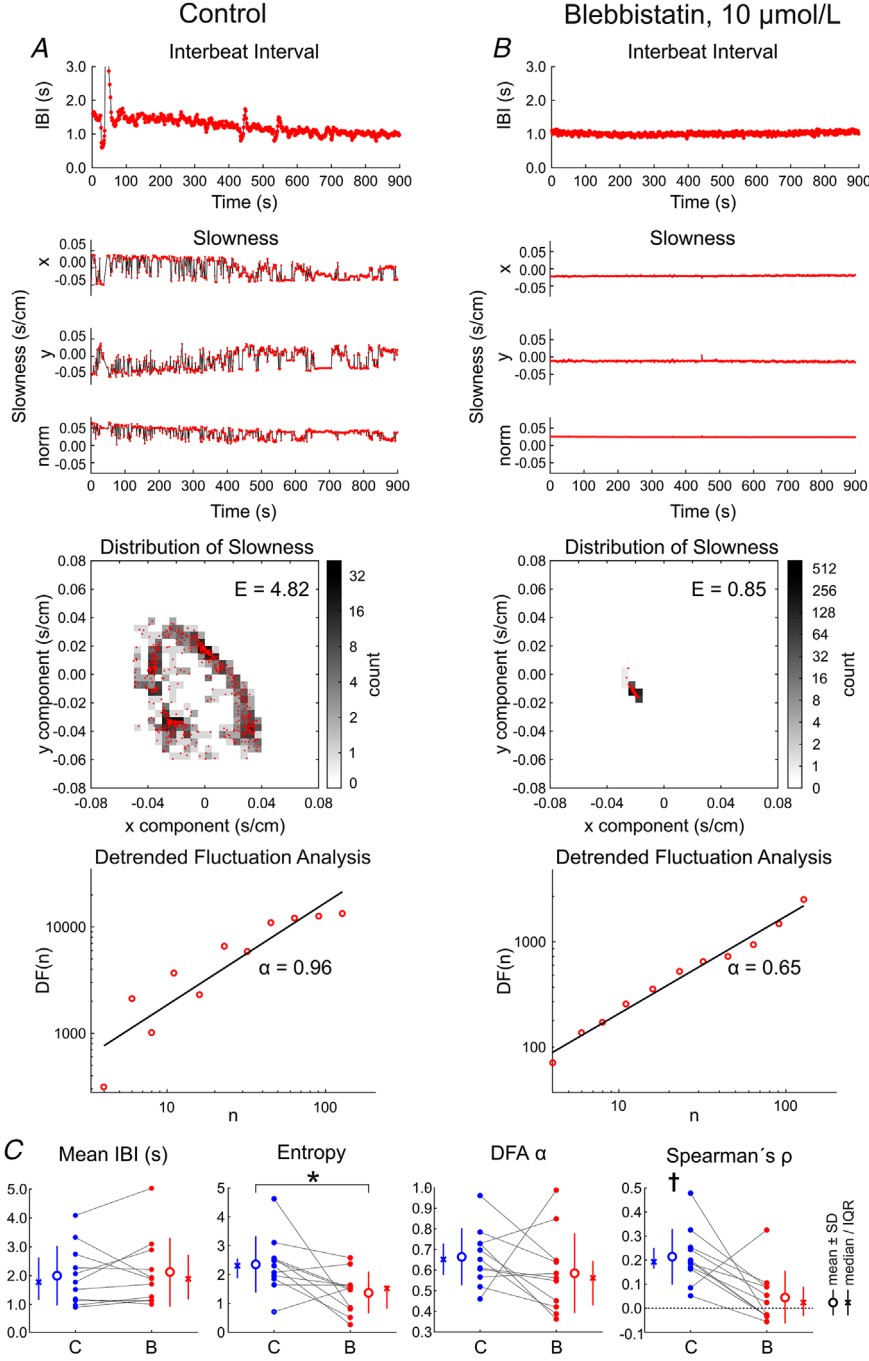

**Figure 3. Effects of 10 $\mu$mol/l blebbistatin on the spatiotemporal characteristics of excitation variability (without external deformation)**
*A*, from top to bottom: interbeat interval (IBI) time series, *x* and *y* components and norm of the slowness vector for every activation, distribution of the tips of the slowness vectors in the *x–y* plane (red dots) with two-dimensional histogram (greyscale), and detrended fluctuation analysis (DFA) plot, for a preparation in control conditions. *B*, same as *A*, for the same preparation after application of blebbistatin. *C*, comparison of mean IBI, entropy of the slowness distribution, DFA exponent *α* and Spearman's *ρ* (same analysis as in Fig. 2*H*) in control conditions (labelled 'C') *versus* with blebbistatin (labelled 'B'). \*$P < 0.05$ (Student's paired *t* test); †$P < 0.05$ to be positive (Student's *t* test). $n = 11$ different monolayer preparations obtained from eight litters.

from zero), indicating that this correlation disappeared. Thus, in summary, blebbistatin decreased the spatial complexity of spontaneous activity and the correlation between activation pattern changes and IBIs.

A series of experiments with the same protocol and analysis was conducted to assess the effects of the non-specific SAC blocker streptomycin. As shown in Fig. 4, streptomycin (20 $\mu$mol/l) exerted no significant effects on mean IBI ($P = 0.273$, Student's paired $t$ test), the entropy of the slowness vector distribution ($P = 0.0766$, Student's paired $t$ test) and the DFA exponent, $\alpha$ ($P = 0.311$, Student's paired $t$ test). Furthermore, Spearman's $\rho$ remained positive not only in control conditions ($P = 0.0138$, Student's one-sample $t$ test *vs.* 0), but also with streptomycin ($P = 0.00662$, Student's one-sample $t$ test *vs.* 0). These results suggest that the contribution of streptomycin-sensitive SACs in determining or influencing beating variability is minor in our preparations.

### Stretch and release both increase beat rate transiently

Next, we aimed at investigating the response of our cultures in terms of spontaneous beat rate upon mechanical stretching and subsequent release. Owing to the background BRV, the examination of the effects of stretch and release required averaging over repeated measurements. Therefore, during 15 min recordings, a predefined level of strain was applied to the preparations and maintained for 1 min; the preparation was then released for 2 min to its original undeformed configuration, and this protocol was repeated five times. Figure 5 illustrates such an experiment and the corresponding analysis for 5% uniaxial strain applied in the $x$ direction (without any pharmacological agent). Figure 5*A* illustrates a typical experiment with the stretch protocol and the corresponding IBI and IBR time series. The IBI series was sampled irregularly, whereas the IBR series was resampled from the IBI at a constant rate of 2 Hz (the IBR corresponds to the reciprocal of IBI).

This approach permitted us to align (i.e. offset) the IBR series on the timings of individual stretches and releases (identified as the onset of linear motor stage action). Given that the stages had a finite speed, it took ~2 s to complete the stretching process until the target strain was obtained, and the same time to return to the initial undeformed configuration.

The top graph in Fig. 5*B* shows raster plots before and after the five stretches. For all stretches, the beat rate increased during the stretch phase (between 0 and 2 s). The aligned IBR data (middle graph) show that IBR increased several-fold during stretch, reaching a maximum at the end of the stretch phase. This increase was transient, however, and the IBR returned to its baseline level within ~8 s. The bottom graph of Fig. 5*B* shows the same data after normalization of the IBR series to their average value over 5 s before the start of stretch (normalized beat rate).

The same analysis was conducted for the five release episodes, as shown in Fig. 5*C*. As visible in the raster plot, release towards the undeformed configuration also increased the beat rate, except in the second release episode. A few seconds before that second episode, the beat rate increased and decreased spontaneously; such excursions of beat rate are typical of BRV (see, e.g. Figs 2*A* and 3*A*). Nevertheless, on average, IBR also increased upon release in a manner similar to that upon stretch (albeit to a lesser extent), and this increase also dissipated after a few seconds.

From every individual monolayer culture, the value of normalized beat rate at the end of the stretch or release phase (when beat rate typically reached its maximum) was extracted for all five stretches and releases, and these values were used for the subsequent summary analysis. The averaged series of normalized beat rate (corresponding to the coloured traces in the bottom graphs of Fig. 5*B* and *C*) were also extracted.

Figure 6 shows this summary analysis for 25 pooled preparations. The analysis included preparations that were later exposed to blebbistatin or streptomycin, because all

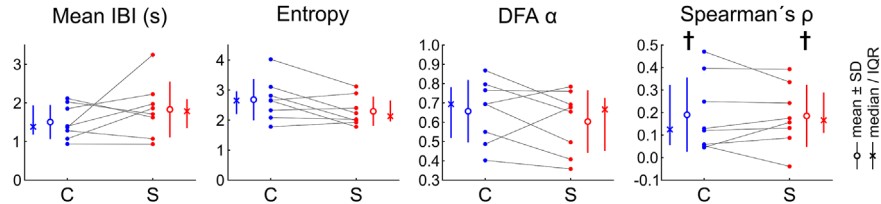

**Figure 4. Summary analysis of spatiotemporal characteristics of excitation variability (without external deformation) for the experiments with streptomycin (20 $\mu$mol/l)**
Comparison of mean interbeat interval (IBI), entropy of the slowness distribution, detrended fluctuation analysis (DFA) exponent $\alpha$ and Spearman's $\rho$ in control conditions (labelled 'C') *versus* with streptomycin (labelled 'S'). $P > 0.05$ for mean IBI, entropy and DFA $\alpha$ (Student's paired $t$ test). †$P < 0.05$ to be positive (Student's $t$ test). $n = 8$ different monolayer preparations obtained from six litters. Same layout as Fig. 3*C*.

these experiments were conducted in the same conditions. Figure 6*A* shows the averaged response to stretch (5% uniaxial strain in the *x* direction) and release in the individual preparations, together with the mean response. This mean response confirms that on average, both stretch and release increased beat rate transiently. However, this response exhibited a large variability between the individual preparations.

Moreover, the baseline average beat rate (left graph of Fig. 6*B*) was also variable, ranging between 0.2 and 2 Hz. To examine whether there was a relationship between baseline beat rate and the magnitude of the response to stretch and release, the preparations were ranked from lowest to highest baseline beat rate (computed as the average IBR before the stretch and release interventions). In the middle and right graphs of Fig. 6*B*, the individual normalized IBR changes for stretch and release are shown according to this ranking. Although most data points were greater than one (representing an increase), there was no significant correlation between average beat rate and the magnitude of the response of IBR to stretch and release.

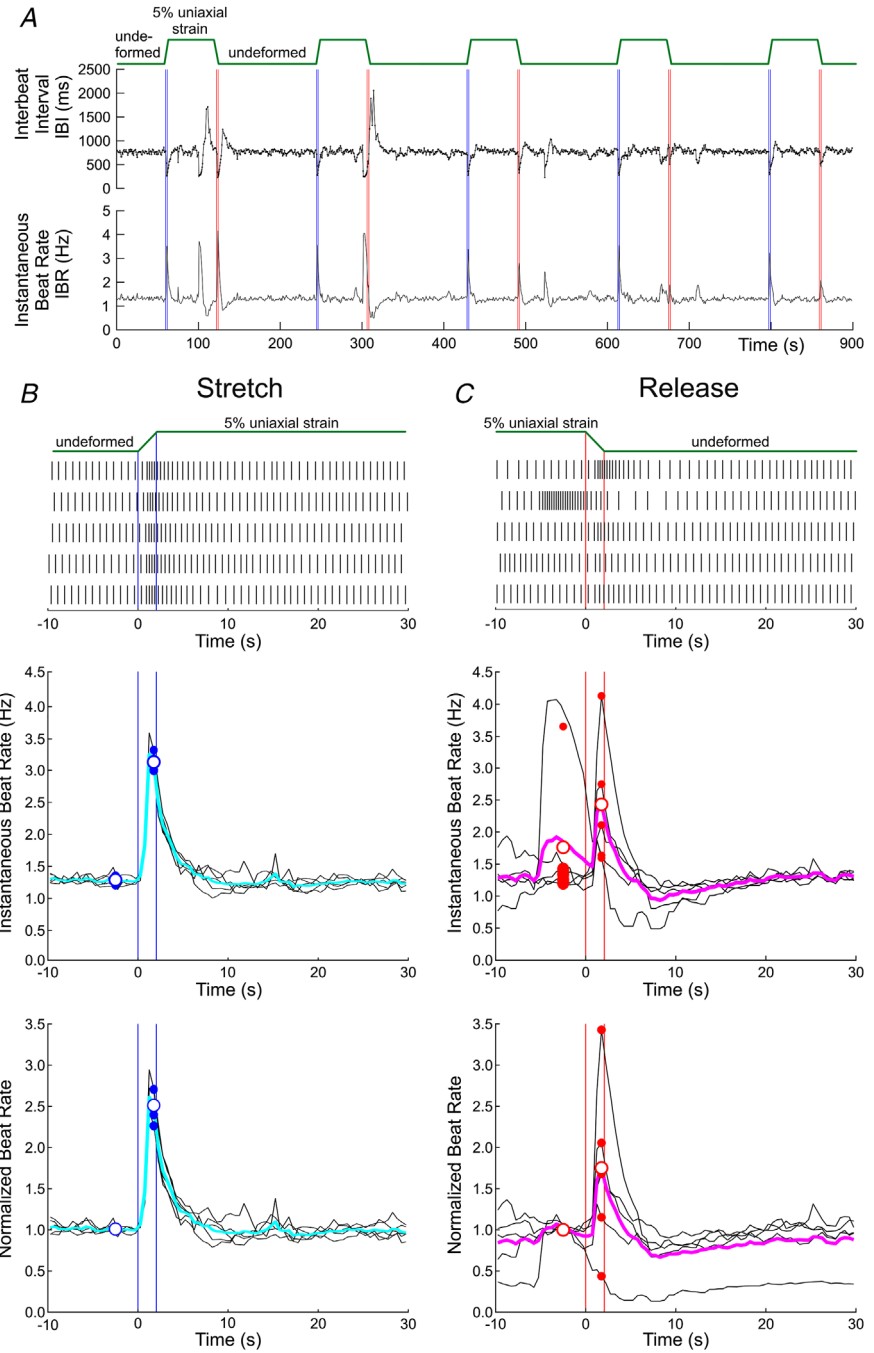

**Figure 5. Spontaneous beat rate in an example preparation subjected repeatedly to 5% uniaxial strain**
*A*, interbeat interval (IBI) series and corresponding instantaneous beat rate (IBR) for a 15 min recording, during which 5% uniaxial strain was applied repeatedly and held for 1 min in the *x* direction (protocol on top). The pairs of vertical lines indicate the start and stop of the linear motors upon stretch (blue) and release (red). *B*, top, raster plot of electrical activations 10 s before and 30 s after each individual stretch (protocol on top), aligned on motor start and end (vertical lines). *B*, middle, corresponding IBR (black, individual stretches; and cyan, average). The small dots at *t* = −2.5 s represent the average beat rate during the last 5 s before stretch, and the open circle is the average of these values. The small dots at *t* = 1.75 s represent the IBR at the end of the stretching phase, and the open circle is the average of these values. *B*, bottom, same IBR data, but normalized to the average IBR during the last 5 s before stretch. Time is offset to zero at motor start. The time scale is identical for all plots. *C*, same analysis and representation as in *B*, but for 10 s before and 30 s after each individual release and with a different colour coding (magenta and red instead of cyan and blue, respectively).

The summary statistical analysis is shown in Fig. 6C. The left graph shows the mean normalized change of IBR upon stretch and release for 5% uniaxial strain in the $x$ direction (the data points correspond to the dots shown in Fig. 6A). The distributions were not normal, and the Wilcoxon signed rank test was therefore used to

test for a difference from one. The median normalized change in beat rate was 2.02 for stretch (IQR, 1.73 to 2.69, $P = 1.23 \times 10^{-5}$, Wilcoxon signed rank test) and 1.42 for release (IQR, 1.03 to 1.76, $P = 0.000174$, Wilcoxon signed rank test). Thus, both stretch and release significantly increased beat rate. The same statistical analysis was

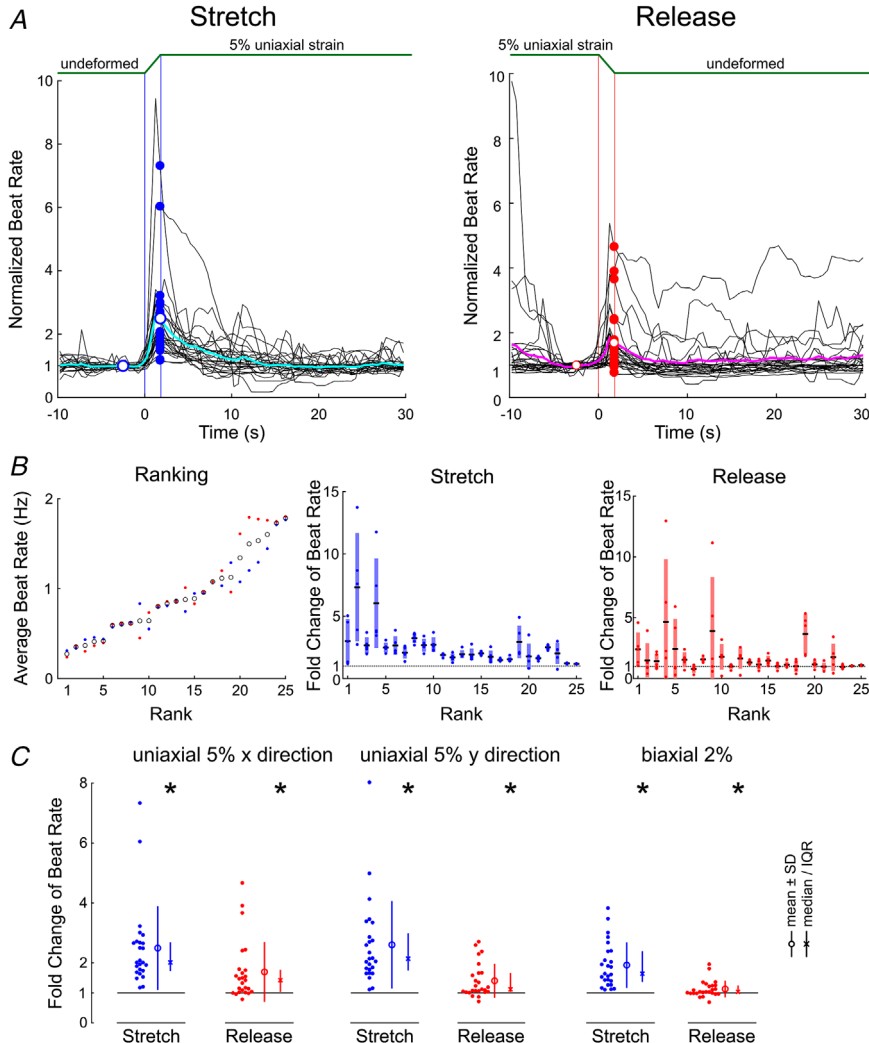

**Figure 6. Effects of stretch and release on spontaneous beat rate (without any pharmacological agent)**
*A*, normalized average beat rate time series for 25 experiments upon stretch (left) and release (right) for uniaxial 5% strain in the *x* direction. The black traces represent the average normalized beat rate series in individual preparations. The coloured traces are the averages for the 25 experiments for stretch and release, respectively. Vertical lines indicate the onset and end of the linear stage movement. The open circles at *t* = −2.5 s represent the average normalized beat rate before stretch and release (defined as one). The coloured dots at *t* = 1.75 s represent the mean normalized (peak) increase in individual preparations, and the open circle marks the mean of all these values. *B*, left, ranking of the experiments according to their average beat rate in the undeformed position. The blue dots are the average beat rates immediately before the stretches, and the red dots are the average beat rates immediately before the releases. The black circles are the averages of these two values and served to rank the preparations. *B*, middle, for the corresponding rank, the graphs report the five normalized increases of beat rate upon stretch (coloured dots), the corresponding means (black horizontal bars) and standard deviations (pale-coloured bars). *B*, right, same analysis as in the middle panel, but for releases. *C*, average normalized (fold) change of instantaneous beat rate (values greater than one indicate an increase) in the 25 preparations (obtained from 16 litters) upon stretch and release, for three different strain protocols. *P < 0.05, Wilcoxon signed rank test.

conducted for 5% uniaxial strain in the $y$ direction (middle graph), whereby the median normalized change in beat rate was 2.12 (IQR, 1.74 to 2.95, $P = 1.23 \times 10^{-5}$, Wilcoxon signed rank test) and 1.11 (IQR, 1.03 to 1.66, $P = 0.000891$, Wilcoxon signed rank test) for stretch and release, respectively. The effect of deformation was similar to that with uniaxial strain in the $x$ direction, which is consistent with the isotropy of the cell cultures and the symmetry of the sMEAs and the stretching system. Finally, as shown in the rightmost graph in Fig. 6C, 2% biaxial strain also increased beat rate upon both stretch (by a factor 1.65, IQR, 1.38 to 2.40, $P = 1.23 \times 10^{-5}$, Wilcoxon signed rank test) and release (by a factor 1.04, IQR, 0.99 to 1.25, $P = 0.0173$, Wilcoxon signed rank test).

## Blebbistatin attenuates the increase of beat rate upon stretch

To investigate whether blebbistatin modulates the increase of beat rate upon stretch and release, we conducted a series of experiments with eight preparations, with which we repeated the same experiments and the same analysis as shown in Fig. 5 after adding 10 $\mu$mol/l blebbistatin to the culture bath. Figure 7A shows the normalized change of beat rate upon stretch and release (5% uniaxial strain in the $x$ direction) in control conditions and with blebbistatin, for an example preparation. On average over the five stretch episodes, the increase of beat rate upon stretch was attenuated by blebbistatin when compared with control conditions. In this preparation, the response of beat rate upon release was also attenuated by blebbistatin. Figure 7B shows the summary analysis for the eight preparations.

Blebbistatin significantly attenuated the normalized change of beat rate upon stretch from $2.30 \pm 0.46$ to $1.57 \pm 0.34$ for 5% uniaxial strain in the $x$ direction ($P = 0.00198$, Student's paired $t$ test), from $2.26 \pm 0.62$ to $1.36 \pm 0.49$ for 5% uniaxial strain in the $y$ direction ($P = 0.0251$, Student's paired $t$ test) and from $1.62 \pm 0.31$ to $1.18 \pm 0.32$ for 2% biaxial strain ($P = 0.0450$, Student's paired $t$ test). Overall, however, blebbistatin did not significantly change the response of beat rate to release ($P = 0.126$, 0.461 and 0.511 for 5% uniaxial strain in the $x$ direction, $y$ direction and 2% biaxial strain, respectively). Of note, blebbistatin could

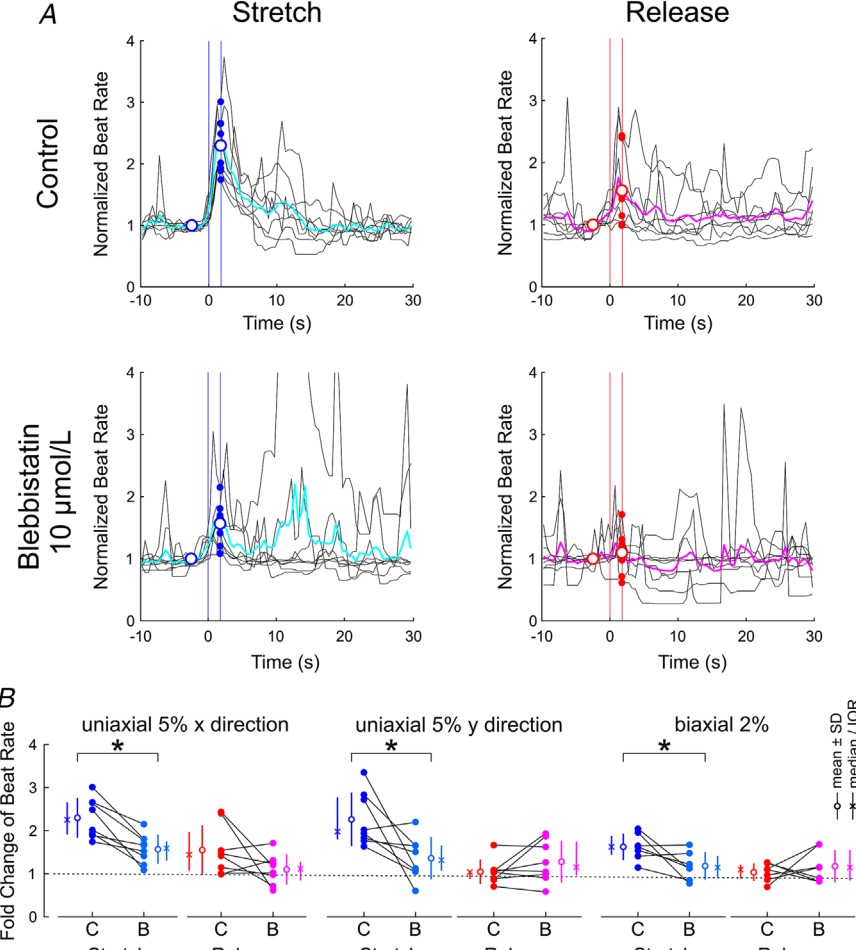

**Figure 7. Effect of 10 $\mu$mol/l blebbistatin on the response of beat rate to stretch and release**
*A*, normalized beat rate in an example preparation upon stretch (left) and release (right) in control conditions (top) and after the addition of blebbistatin (bottom); 5% uniaxial strain in the $x$ direction. Same analysis as in Fig. 5. *B*, comparison of the normalized (fold) change in beat rate upon stretch and release in control conditions (labelled 'C') *versus* with blebbistatin (labelled 'B'), for the three different target strains. *$P < 0.05$, Student's paired $t$ test. $n = 8$ monolayer preparations obtained from seven litters, except for the 2% biaxial strain protocol ($n = 7$).

not be washed out (as confirmed after several washouts by the persistent absence of contractions under microscope inspection; see Supplementary Video S1), and it was thus not possible to assess whether the effect of blebbistatin was reversible.

Finally, taking the same approach, we examined in a series with seven preparations whether streptomycin also modulates the response of beat rate to stretch. As shown in Fig. 8 for an example experiment (Fig. 8*A*) and in the summary analysis (Fig. 8*B*), streptomycin did not significantly modify the increase of beat rate upon stretch ($P = 0.578$, 0.367 and 0.844 for 5% uniaxial strain in the *x* direction, *y* direction and 2% biaxial strain, respectively) and release [$P = 0.297$, 0.313 and 0.563 for 5% uniaxial strain in the *x* direction ($n = 7$), *y* direction ($n = 6$) and 2% biaxial strain ($n = 6$), respectively]. These results thus indicate that the response of beat rate to deformation is modulated by blebbistatin, but that the effect of streptomycin, if any, is minor and undetectable in our experiments.

## Discussion

Our main findings can be summarized as follows. In experiments without stretch, we observed that spontaneous electrical activity in our monolayer cultures (8 mm in diameter) originates from the periphery of the preparations, with a site of origin prone to abrupt and unpredictable changes. This recapitulates our previous findings (Ponard et al., 2007) in addition to the observations of others (Boudreau-Béland et al., 2015). The DFA exponent was, on average, near 0.65, confirming the presence of long-range temporal correlations and a power-law behaviour of BRV, as we described previously (Kucera et al., 2000; Ponard et al., 2007). The fact that the DFA exponent was less than one suggests that *in vivo*, further factors are likely to be involved in shaping HRV. No change of this exponent was detected upon the application of either blebbistatin or streptomycin, suggesting that the fractal behaviour of BRV is not conditioned by active force generation or SACs. However, blebbistatin decreased the complexity of the spatial

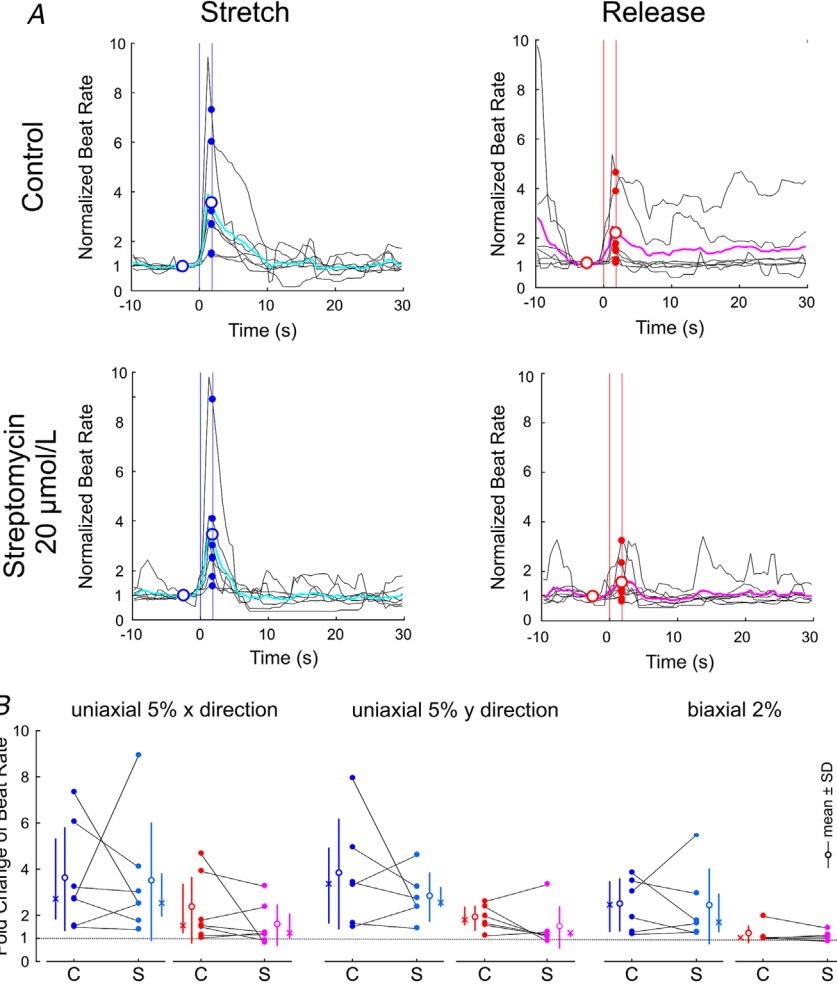

**Figure 8. Effect 20 $\mu$mol/l streptomycin on the response of beat rate to stretch and release**

*A*, normalized beat rate in an example preparation upon stretch (left) and release (right) in control conditions (top) and after the addition of streptomycin (bottom); 5% uniaxial strain in the *x* direction. Same analysis as in Figs 5 and 7. *B*, comparison of the normalized (fold) change in beat rate upon stretch and release in control conditions (labelled 'C') *versus* with streptomycin (labelled 'S'), for the three different strain protocols.

$n = 6$–7 monolayer preparations obtained from five litters. $P > 0.05$ for all comparisons (Student's paired *t* test).

variability of activation patterns, suggesting that active force generation is involved mechanistically in this variability. Moreover, blebbistatin disrupted the positive correlation between IBIs and the interbeat slowness differences.

In experiments with stretches and releases, repetitions of the stretch–release protocols and averaging demonstrated that stretch increases beat rate. At a first glance, this result appears logical, being in line with previous studies (Cooper & Kohl, 2005; Kohl et al., 1994) and with the notion that SACs potentiate depolarization during diastole (Riemer & Tung, 2003). However, streptomycin had no influence on the response to stretch, whereas blebbistatin, in contrast, attenuated this response. Furthermore, this increase in beat rate was not sustained but transient and dissipated after a few seconds, suggesting inactivation or desensitization of the stretch-sensing mechanism. In addition, releasing the preparations from strain also increased the beat rate. This result is unexpected, because intuitively, one would anticipate that application of the reverse mechanical intervention (release) would cause the opposite effect (slowing of beating), or no effect at all if the stretch-sensing mechanism is desensitized.

Based on experiments in volume-loaded ventricles and computational analyses, Mills et al. (2008) suggested that stretch could lead to an increase in membrane capacitance. The subsequent work of Pfeiffer et al. (2014) indicated that stretch increases the capacitance of cultured ventricular myocytes via recruitment of caveolae to the sarcolemma. Assuming that the number and function of membrane channels and transporters is not affected by stretch, the same ion currents would then have to charge/discharge a larger capacitive load, which is expected to slow both depolarization and repolarization, and thus spontaneous beat rate. However, we did not observe this in our study. Given that the time course of caveolae incorporation and possible capacitance changes is unknown, we can, nevertheless, speculate that an increase in capacitance could have contributed to the slow return of beat rate towards baseline after a few seconds. Further studies with dedicated systems, such as the platform developed by de Coulon et al. (2021), might provide an appropriate answer in the future.

The mechanisms by which cells sense mechanical cues are exquisitely complex. According to Cox et al. (2019), ion channels can sense stretch directly, via membrane phospholipids (force-from-lipids principle), or indirectly, via cytoskeletal proteins or extracellular tethers with which the channels interact (force-from-filaments principle). In the context of our findings, it appears more plausible (if ion channels are involved) that the influence of active force generation is mediated by the force-from-filaments principle. In addition, as reviewed by Prosser and Ward (2014) and by Boycott et al. (2020),

mechanical influences can lead to intracellular signalling cascades, leading to modulation of ion channel function by second messengers, such as NO or extracellular signal-regulated kinase (ERK). More specifically, NO, produced in cardiomyocytes by both endothelial and neuronal NO synthases, is influenced by mechanical forces. In turn, NO impacts the function of integrins, which link the actin cytoskeleton to the extracellular matrix. Nitric oxide also influences other cytoskeletal proteins (e.g. talin, vinculin) that are able to sense and integrate mechanical signals. It was suggested that NO stimulation of integrins then promotes NO-mediated $Ca^{2+}$ release from the sarcoplasmic reticulum and modulates the function of L-type $Ca^{2+}$ channels (Boycott et al., 2020). In the context of pacemaker function, this would exert a repercussion on the $Ca^{2+}$ clock and affect beat rate. To investigate the involvement of NO in mediating the effects of active force on the response to stretch, experiments could be performed in which NO production is depressed by the NO synthase inhibitor L-NAME or enhanced by the NO donor SNAP (Boycott et al., 2020), in the presence *versus* absence of blebbistatin.

These diverse mechanisms might react not only to passive tension, but also to active forces generated by the cellular contractile apparatus. These intricate mechano-sensing mechanisms are likely to operate jointly, and this complex picture must be taken into account in the interpretation of our results. In the following, we will present and discuss a few hypotheses to explain the surprising, unexpected and sometimes paradoxical findings of our study.

## Active force generation and the spatiotemporal patterns of pacemaker activity

Let us consider a network of pacemaker cells connected electrically by gap junctions and assume that individual cells do not have the same intrinsic rate (i.e. the rate they would exhibit if they were disconnected from their neighbours). When coupled, the network synchronizes by mutual entrainment, and the activity then originates from a cell or a region of cells having the highest rate (Aghighi & Comtois, 2017; Gratz et al., 2020; Jalife, 1984; Kanakov et al., 2007; Ponard et al., 2007). This region, by initiating propagated action potentials, then drives the rest of the tissue. Consider now the scenario in which this primary pacemaker region exhibits a decrease of its intrinsic rate (owing to intrinsic BRV or, in the intact organism, owing to a shift of the sympathovagal balance). In such a situation, another region will take over the primary pacemaking role. This situation arises naturally in the heart when, for example, the atrioventricular node takes over the primary pacemaker role during extreme sinus node bradycardia. In our cultures, this change of

pacemaker focus is reflected by a large change in the slowness vector. Conversely, when the primary pacemaker region exhibits an increase of its intrinsic rate, this region will remain the primary pacemaking site, and no large change in slowness will be observed. Hence, these considerations explain the positive correlation that we observed between IBIs and interbeat slowness differences.

Curiously, in our experiments, this correlation was disrupted by blebbistatin, indicating that active force generation is involved in the underlying mechanism. How do active contractions contribute to this phenomenon? Nitsan et al. (2016) have shown that the spontaneous beating of isolated cardiomyocytes cultured on poly-acrylamide gel can be entrained and synchronized to the oscillating movement of a microprobe inserted into the gel at a short distance from the cell, but not touching it. Remarkably, even after stopping this mechanical stimulation, the myocytes maintained their spontaneous rate close to the rate of the probe movement for up to 1 h, indicating that the cells underwent a long-lasting conditioning that permitted them to 'learn' this new rate (Nitsan et al., 2016). This entrainment was inhibited by the contraction uncouplers butanedione monoxime and blebbistatin, and by the microtubule polymerization inhibitor colchicine, but not by the SAC blockers gadolinium and GsMTx-4. This entrainment was also observed for two cells located close to each other, but without physical contact (Nitsan et al., 2016; Viner et al., 2019). As mechanotransduction pathways, the authors proposed a concerted action between microtubules, NO synthase 2, reactive oxygen species, calmodulin kinase II and ryanodine receptors, which would, ultimately, regulate calcium release and the calcium clock (Prosser & Ward, 2014; Viner et al., 2019).

Based on these findings, we postulate that active rhythmic contractions in our preparations also condition the myocytes to develop an intrinsic rate close to that of the primary pacemaking site. Such conditioning would not be mediated by direct mechanical communication via the growth substrate (because PDMS is orders of magnitude stiffer than polyacrylamide gel) or by mechanical cell–cell junctions, but by the contraction wave accompanying every action potential. This would permit conditioning over large distances. Hence, when the primary pacemaking site fails (i.e. its rate decreases), another region, at a potentially remote location, can take over immediately. This mechanism would then explain the correlation between IBIs and slowness differences seen in our experiments and the complex spatial variation of the activation pattern. It would also explain why this correlation and this spatial complexity are decreased by blebbistatin. In the presence of blebbistatin, without active contractions, it becomes less likely that a remote site will take over the primary pacemaker activity, and the number of such sites will be decreased, as observed in our

experiments. In this manner, active cellular contractions might contribute to the regulation of spontaneous activity in both the temporal and spatial domains.

In short-term optical mapping experiments using a calcium-sensitive dye, Boudreau-Béland et al. (2015) also observed frequent shifts of the pacing focus in neonatal rat ventricular myocyte monolayer cultures grown on PDMS. Upon adrenergic stimulation with isoprenaline, beat rate increased, whereas the number of pacing sites decreased (less spatial variability). On the one hand, our observation that short IBIs are associated with smaller changes in the activation pattern is in line with this finding. On the other hand, isoprenaline, by its inotropic effect, should reinforce the involvement of active contractions and augment the spatial variability of the pacemaking site. Hence, we surmise that the mechanism mediated by active contractions is already activated fully, even without additional adrenergic stimulation. Untangling these effects (which could be achieved by experiments with both isoprenaline and blebbistatin) was, however, not our primary goal.

## Active force generation and the response of beat rate to stretch

A finding of interest is that the acceleration of beating upon stretch was not sustained, but dissipated after a few seconds. This observation is consistent with the work of Cooper and Kohl (2005), who also reported a transient increase in the beat rate of guinea-pig sino-atrial tissue. Thus, it appears that the underlying mechanism desensitizes. Curiously, stretching of murine SAN strips caused a decrease in beat rate, and the authors explained this species difference by the different repolarization dynamics (Cooper & Kohl, 2005).

Importantly, in this same work (Cooper & Kohl, 2005), streptomycin did not influence the response of beat rate to stretch, and our results are in line with these observations. In contrast, blebbistatin clearly attenuated the stretch-induced increase of beat rate in our experiments. This supports the notion that the modulation of spontaneous beat rate by stretch is not primarily mediated by channels directly activated by stretch (or at least by the family of such channels that are sensitive to streptomycin). Rather, our results lend support to the hypothesis that beat rate modulation is mediated by mechano-sensing mechanisms of another type, possibly via the force-from-filaments principle, and that this mechanism depends on the active contraction generated by actin–myosin filaments. Irrespective of the mechano-sensing pathway, the function of membrane channels must, in the end, be altered in favour of depolarizing currents to produce an acceleration of beating. This could happen via modulation of other ion channels, calcium signalling or NO synthesis in the

cardiomyocytes. Calcium signalling and NO have recently captured the attention of scientists with their contribution to MEF (Boycott et al., 2020; Prosser & Ward, 2014).

### Why does release also increase beat rate?

Another interesting result was that release also increased beat rate, for which we propose the following explanation. Assuming incompressibility, stretching a three-dimensional object in one direction is accompanied necessarily by a constriction in another direction. In experiments with, e.g. 5% uniaxial strain, our stretching set-up compensates the constriction in the $y$ direction in the plane of the culture substrate. Thus, the substrate area is increased by 5%. This implies that in the $z$ direction (normal to the substrate), the cells are constricted by 5%. A similar consideration applies for 2% equibiaxial strain, whereby the area increases by ~4%, and thus a constriction of ~4% occurs in the $z$ direction. Upon release, the opposite happens, which implies a stretch in the $z$ direction. Although cultured cardiomyocytes typically flatten out on the substrate (Jousset et al., 2016), it is, nevertheless, plausible that another (probably smaller) pool of mechanosensitive proteins could react to this deformation along the $z$-axis upon release. Thus, an initial pool might react to stretch before desensitizing, followed by the activation of the other, not yet desensitized, smaller pool upon release. Therefore, *a fortiori*, any change in the shape of the cultured tissue might increase beat rate. These considerations also explain why, on average, the increase in beat rate was smaller (but still statistically significant) upon stretch when compared with release.

However, it cannot be excluded that two completely different mechano-sensing mechanisms might be involved upon stretch and upon release. We did not detect any significant difference of the response of beat rate to release before and after the application of blebbistatin. Although the reason for this might be a smaller effect size precluding its detection, it is also possible that two different mechano-sensing pools might have been involved, one modulated by active contraction and the other not.

### Is it possible that myofibroblasts caused the observed effects?

In intact cardiac tissue, fibroblasts and myofibroblasts outnumber myocytes and therefore represent an important myocardial component (Quinn & Kohl, 2021; Rohr, 2009). Earlier work showed that fibroblasts can form connections with myocytes (Quinn et al., 2016), and heterocellular communication between myocytes and myofibroblasts in cell cultures is well established

(Gaudesius et al., 2003; Miragoli et al., 2006, 2007). Cardiomyocytes can even form connections with other cell types, such as macrophages (Hulsmans et al., 2017). Although we minimized the number of myofibroblasts by preplating the dissociated cells and limited their proliferation with bromodeoxyuridine, we cannot exclude the presence of a residual number of such cells, which would influence the automaticity in our cultures in control conditions and especially upon stretch. Recently, de Coulon et al. (2021) reported large stretch-activated inward currents in NIH 3T3 cells, a cell line derived from fibroblasts. Thus, it is possible that the responses to deformation that we observed might have been caused by myofibroblasts or other cell types.

In co-cultures of myocytes and myofibroblasts, Grand et al. (2014) observed that streptomycin decreases the depolarizing effect of myofibroblasts on myocytes owing to constitutively active SACs expressed in myofibroblasts. Based on this, if the proportion of myofibroblasts was substantial in our preparations, we would expect that streptomycin would decrease the spontaneous beat rate of our cultures even in the absence of stretch and attenuate the increase of beat rate upon stretch. Neither was observed, suggesting that the role of myofibroblasts in our preparations was minor, if not absent. Of note, cardiac myofibroblasts in culture systems require the presence of serum to survive, and experiments with such cells are typically conducted with 1% neonatal calf serum (Grand et al., 2014; Miragoli et al., 2007; Thompson et al., 2011). Our recordings were conducted in the absence of serum, which minimizes the possibility that myofibroblasts might have affected our observations.

### Clinical implications

In the future, biological pacemakers based on genetically engineered cardiac cells or stem cell-derived cardiomyocytes might serve as an alternative to electronic pacemakers for patients suffering from sinus node bradycardia (Rosen et al., 2011). In addition, notable progress has been made in cardiac tissue bioengineering for drug testing and regenerative purposes, e.g. patches to graft after myocardial infarction (Weinberger et al., 2017). It is thus important to understand how these types of engineered cells and tissues behave in terms of their propensity to generate normal pacemaker activity (if this is their primary goal), but also ectopic or pro-arrhythmic activity (in the case of, e.g. ventricular patches). Such engineered tissues might increase their rate of spontaneous activity upon stretch. This might be beneficial in the case of pacemaker tissue, but also deleterious in the case of ventricular tissue constructs, because it might potentiate extrasystoles and thus re-entrant arrhythmias during increased preload. Our

results also suggest that the active contractile properties of normal, diseased or engineered myocardium might determine its response to mechanical deformation, an aspect that deserves to be investigated further.

## Limitations

It might be argued that contracting elements are scarce or absent in the centre of the SAN and that the role of active force generation is not relevant in that region. However, as shown in the study by Bychkov et al. (2020), the SAN consists of highly intertwined F-actin-positive and F-actin-negative tissue. Thus, contractile elements are not completely absent in the centre of the SA node. Moreover, both active contractile and passive elastic forces can be transmitted over a certain distance (several cells) owing to the action–reaction principle (Newton's third law). It is possible that the role of active force generation is more prominent at the periphery of the SAN or at its interface to the atria. It might also be argued that the expression of Cx43 and sodium channels is very low in the SAN centre, where the impulse is initiated, in contrast to our monolayer preparations, and this represents a limitation of our culture model in this regard. Ideally, one should conduct a similar study with intact SAN tissue, preferably with SANs of larger mammals. The challenges that we foresee will be to ensure that the contact between the tissue and the sMEA remains tight and that the tissue closely follows the movement of the sMEA upon stretch. These challenges will require further developments in bioengineering.

In our study, we used blebbistatin to block active force generation. Nonetheless, it cannot be excluded that blebbistatin exerts non-specific effects, either directly on ion channels or $Ca^{2+}$-handling proteins, or indirectly by decreasing ATP consumption and the cellular metabolic load. Increased ATP availability might potentiate the function of sarcolemmal and sarcoplasmic pumps and inhibit the ATP-sensitive potassium current (Swift et al., 2021). Decreased metabolic load might modify the function of mitochondria, which are organelles that participate in cardiomyocyte $Ca^{2+}$ homeostasis (Shannon & Bers, 2004). These considerations about metabolism will apply to any contraction uncoupler.

In our study, we used streptomycin, which might also block potassium channels and L-type $Ca^{2+}$ channels (Quinn & Kohl, 2021). Based on our previous observation that blocking L-type $Ca^{2+}$ channels with nifedipine completely suppresses spontaneous electrical activity in cardiomyocyte cultures (Ponard et al., 2007), such a non-specific effect would probably have depressed beat rate in our preparations, which we did not observe. There might also be conditions for which the expression of SACs could change, e.g. by streptomycin exposure during the culture period preceding the experiments, which

might contribute to the absence of streptomycin effects. However, we found no specific evidence in the literature that streptomycin up- or downregulates the expression of SACs.

Our stretching system allowed only slow stretches (∼2 s to reach target strain), with a speed that is lower compared with the strain rates present during the normal heart beat. Furthermore, our system did not allow a cyclic stretch and relaxation pattern synchronized with the electrical activity. Nevertheless, our stretch patterns mimicked an acute increase in preload, as would typically occur *in vivo* during any situation with an increased end-diastolic volume, such as increased venous return. Of note, our system permitted accurate control and quantification of the applied strain, thus removing the variability of this factor between individual experiments.

Although more spatial detail would have been obtained with optical mapping using a voltage-sensitive dye, our use of sMEAs circumvented the problem posed by the long-term phototoxic damage inherent to such dyes. Moreover, at present, available optogenetic voltage reporters are still too slow to permit identification of activation times with submillisecond precision, an aspect that was essential in our work.

It was also not possible with our system to apply strains >5% (uniaxial) or >2% (biaxial) owing to the frequent loss of electrical conductivity of the gold interconnectors at these strain levels. Thus, examination of the effects of larger strains was not possible. Nevertheless, it is remarkable that such low levels of deformation already induce measurable changes in beat rate.

Finally, it remains unknown to what extent our results can be extrapolated and translated to the intact heart and the human heart. Our platform, permitting the simultaneous control of strain and recording of electrical activity, might, nevertheless, open avenues with other types of cardiac preparations in the future.

## Conclusion

Our results indicate that active force generation contributes mechanistically to the complexity of spatiotemporal excitation patterns of spontaneous pacemaker activity. In addition, both stretch and release, hence any change of shape, can accelerate beating. Our study points to cellular phenomena operating separately from ion channels directly activated by stretch. Our study thus contributes to the understanding of how MEF might influence beat rate and HRV and suggests that the active contractile properties of normal, diseased or engineered myocardium are essential players in MEF. Recent research has challenged the view that a dominant centre in the SAN drives pacemaking (Clancy & Santana, 2020), suggested that the SAN operates as a critical system

(Weiss & Qu, 2020) and raised the question of whether factors others than gap junctional coupling are involved in impulse formation in the SAN (Weiss & Qu, 2020). Thus, our results suggest that mechanical mechanisms represent a path worthy of exploration.

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

## Additional information

### Open research badges

This article has earned an Open Data badge for making publicly available the digitally-shareable data necessary to reproduce the reported results. The data is available at https://doi.org/10.5281/zenodo.6524580.

### Data availability statement

All raw recordings, the MATLAB source code of the user interface and scripts for generating the panels of Figs 2–8 are openly available on the repository Zenodo (https://doi.org/10.5281/zenodo.6524580).

### Competing interests

The authors have no competing interests to disclose.

### Author contributions

The experiments were performed at the Department of Physiology of the University of Bern. J.P.K. designed the study. S.N. conducted the experiments and prepared the figures. S.N. and J.P.K. analysed the data. S.N., S.P.L. and J.P.K. drafted the manuscript. All authors approved the final version of the manuscript and agree to be accountable for all aspects of the work in ensuring that questions related to the accuracy or integrity of any part of the work are appropriately investigated and resolved. All persons designated as authors qualify for authorship, and all those who qualify for authorship are listed.

### Funding

This work was supported by the Swiss National Science Foundation (grant no. 310030_184707 to J.P.K.).

### Acknowledgements

The authors are grateful to Helene Hinnen for her assistance with the cardiomyocyte cultures, to Michael Känzig for taking care of our animals, to Michael Stoeckel, Anthony Guillet and the entire cleanroom staff at Geneva Campus Biotech for their training, support and advice, to Ange Maguy for his suggestions on statistics and for proofreading our manuscript, and to Lucilla Giammarino for her support with the video recording.

### Keywords

beat rate variability, blebbistatin, cardiac cell culture, heart rate variability, mechano-electrical feedback, pacemaker function, streptomycin, stretch-activated channel, stretchable micro-electrode array

## Supporting information

Additional supporting information can be found online in the Supporting Information section at the end of the HTML view of the article. Supporting information files available:

**Statistical Summary Document**
**Peer Review History**
**Video S1**. Video recording of a monolayer preparation grown on polydimethylsiloxane (PDMS) in control medium (Hanks' balanced salts solution), after application of blebbistatin and upon washout. The video shows that blebbistatin blocks the contractions of the myocytes fully and that the effect of blebbistatin cannot be washed out.

## Translational perspective

Cardiomyocyte cultures are spontaneously active and thus represent a model of a natural cardiac pacemaker. In such cultures, beat rate variability exhibits features similar to those of heart rate variability *in vivo*. It is, however, not fully elucidated how mechano-electrical feedback affects beating variability in such preparations. Using stretchable microelectrode arrays, we investigated how the myosin inhibitor blebbistatin and the non-selective stretch-activated channel blocker streptomycin affect beating variability and the response of the beat rate of these cultures to stretch. We found that blebbistatin decreases the spatial complexity of spontaneous electrical activity and attenuates the increase of beat rate caused by stretch. Interestingly, streptomycin exerted no manifest effects. This supports the notion that active force generation, rather than stretch-activated channels, is involved mechanistically in the complexity of spontaneous beating patterns and in the stretch-induced acceleration of beating. In the future, biological pacemakers based on genetically engineered cardiac cells or stem cell-derived cardiomyocytes might serve as an alternative to electronic pacemakers. Moreover, cardiac tissue bioengineering for drug testing and regenerative purposes is constantly progressing. In this context, our study contributes to the understanding of how these types of engineered cells and tissues behave in terms of their propensity to generate normal pacemaker activity, but also ectopic or pro-arrhythmic activity. Importantly, our results suggest that the active contractile properties of normal, diseased or engineered myocardium might determine its response to mechanical deformation. Finally, our study also suggests that active force generation might influence heart rate variability.

