## [Peer Review History · The Journal of Physiology]

Active force generation contributes to the complexity of spontaneous activity and to the response to stretch of murine cardiomyocyte cultures

Seyma Nayir, Stéphanie P Lacour, and Jan P Kucera

DOI: 10.1113/JP283083

Corresponding author(s): Jan Kucera (kucera@pyl.unibe.ch)

The following individual(s) involved in review of this submission have agreed to reveal their identity: Victor Maltsev (Referee #1); Philippe Comtois (Referee #2)

Review Timeline:

Submission Date:	10-Mar-2022
Editorial Decision:	20-Apr-2022
Revision Received:	17-May-2022
Accepted:	01-Jun-2022

Senior Editor: Bjorn Knollmann

Reviewing Editor: Michael Shattock

Transaction Report:

Dear Professor Kucera,

Re: JP-RP-2022-283083 "Active force generation contributes to the complexity of spontaneous activity and to the response to stretch of murine cardiomyocyte cultures" by Seyma Nayir, Stéphanie P Lacour, and Jan P Kucera

Thank you for submitting your manuscript to The Journal of Physiology. It has been assessed by a Reviewing Editor and by 2 expert Referees and I am pleased to tell you that it is considered to be acceptable for publication following satisfactory revision.

The reports are copied at the end of this email. Please address all of the points and incorporate all requested revisions, or explain in your Response to Referees why a change has not been made.

NEW POLICY: In order to improve the transparency of its peer review process The Journal of Physiology publishes online as supporting information the peer review history of all articles accepted for publication. Readers will have access to decision letters, including all Editors' comments and referee reports, for each version of the manuscript and any author responses to peer review comments. Referees can decide whether or not they wish to be named on the peer review history document.

Authors are asked to use The Journal's premium BioRender (<https://biorender.com/>) account to create/redraw their Abstract Figures. Information on how to access The Journal's premium BioRender account is here:

<https://physoc.onlinelibrary.wiley.com/journal/14697793/biorender-access> and authors are expected to use this service. This will enable Authors to download high-resolution versions of their figures. The link provided should only be used for the purposes of this submission. Authors will be charged for figures created on this premium BioRender account if they are not related to this manuscript submission.

I hope you will find the comments helpful and have no difficulty returning your revisions within 4 weeks.

Your revised manuscript should be submitted online using the links in Author Tasks: Link Not Available.

Any image files uploaded with the previous version are retained on the system. Please ensure you replace or remove all files that have been revised.

REVISION CHECKLIST:

- Article file, including any tables and figure legends, must be in an editable format (eg Word)
- Abstract figure file (see above)
- Statistical Summary Document
- Upload each figure as a separate high quality file
- Upload a full Response to Referees, including a response to any Senior and Reviewing Editor Comments;
- Upload a copy of the manuscript with the changes highlighted.

- A potential 'Cover Art' file for consideration as the Issue's cover image;
- Appropriate Supporting Information (Video, audio or data set https://jp.msubmit.net/cgi-bin/main.plex?form_type=display_requirements#supp).

To create your 'Response to Referees' copy all the reports, including any comments from the Senior and Reviewing Editors, into a Word, or similar, file and respond to each point in colour or CAPITALS and upload this when you submit your revision.

I look forward to receiving your revised submission.

If you have any queries please reply to this email and staff will be happy to assist.

Yours sincerely,

Bjorn Knollmann
Senior Editor
The Journal of Physiology

REQUIRED ITEMS:

- Author photo and profile. First (or joint first) authors are asked to provide a short biography (no more than 100 words for one author or 150 words in total for joint first authors) and a portrait photograph. These should be uploaded and clearly labelled with the revised version of the manuscript. See Information for Authors for further details.
- You must start the Methods section with a paragraph headed Ethical Approval. A detailed explanation of journal policy and regulations on animal experimentation is given in Principles and standards for reporting animal experiments in The Journal of Physiology and Experimental Physiology by David Grundy *J Physiol*, 593: 2547-2549. doi:10.1113/JP270818.). A checklist outlining these requirements and detailing the information that must be provided in the paper can be found at: <https://physoc.onlinelibrary.wiley.com/hub/animal-experiments>. Authors should confirm in their Methods section that their experiments were carried out according to the guidelines laid down by their institution's animal welfare committee, and conform to the principles and regulations as described in the Editorial by Grundy (2015). The Methods section must contain details of the anaesthetic regime: anaesthetic used, dose and route of administration and method of killing the experimental animals.
- Please ensure that the Article File you upload is a Word file.
- A Statistical Summary Document, summarising the statistics presented in the manuscript, is required upon revision. It must be on the Journal's template, which can be downloaded from the link in the Statistical Summary Document section here: https://jp.msubmit.net/cgi-bin/main.plex?form_type=display_requirements#statistics.
- Papers must comply with the Statistics Policy: https://jp.msubmit.net/cgi-bin/main.plex?form_type=display_requirements#statistics.

In summary:

- If $n \leq 30$, all data points must be plotted in the figure in a way that reveals their range and distribution. A bar graph with data points overlaid, a box and whisker plot or a violin plot (preferably with data points included) are acceptable formats.
- If $n > 30$, then the entire raw dataset must be made available either as supporting information, or hosted on a not-for-profit repository e.g. FigShare, with access details provided in the manuscript.
- 'n' clearly defined (e.g. x cells from y slices in z animals) in the Methods. Authors should be mindful of pseudoreplication.
- All relevant 'n' values must be clearly stated in the main text, figures and tables, and the Statistical Summary Document (required upon revision).
- The most appropriate summary statistic (e.g. mean or median and standard deviation) must be used. Standard Error of the Mean (SEM) alone is not permitted.
- Exact p values must be stated. Authors must not use 'greater than' or 'less than'. Exact p values must be stated to three significant figures even when 'no statistical significance' is claimed.
- Statistics Summary Document completed appropriately upon revision.
- Please include an Abstract Figure. The Abstract Figure is a piece of artwork designed to give readers an immediate understanding of the research and should summarise the main conclusions. If possible, the image should be easily 'readable' from left to right or top to bottom. It should show the physiological relevance of the manuscript so readers can assess the importance and content of its findings. Abstract Figures should not merely recapitulate other figures in the manuscript. Please try to keep the diagram as simple as possible and without superfluous information that may distract from the main conclusion(s). Abstract Figures must be provided by authors no later than the revised manuscript stage and should be uploaded as a separate file during online submission labelled as File Type 'Abstract Figure'. Please ensure that you include the figure legend in the main article file. All Abstract Figures should be created using BioRender. Authors should use The Journal's premium BioRender account to export high-resolution images. Details on how to use and access the premium account are included as part of this email.

EDITOR COMMENTS

Reviewing Editor:

Both reviewers found the manuscript interesting but both highlighted issues that need addressing before potential reconsideration. The importance of these results for normal physiology and/or pathophysiology also needs clarifying in the revised Discussion.

Senior Editor:

I concur with the reviewing editor. In your revision, please also consider that blebbistatin will reduce ATP consumption and hence the metabolic load on the cells. This effect should be discussed.

REFEREE COMMENTS

Referee #1:

Using stretchable microelectrode arrays, Nayir et al. investigates complexity of spontaneous activity in murine cardiomyocyte cultures with respect to stretch. The authors apply pharmacological tools to explore two mechanisms that could modulate the activity: active force generation and stretch. The paper is interesting, original, and technically well done. However, the importance of these results for physiology remains unclear to me. I also have several questions about their methods, robustness of the data, and data interpretation.

Major issues.

1) Introduction. You wrote: "Pacemaker cells synchronize via gap junctional coupling by mutual entrainment, whereby the cells or a group of cells exhibiting the fastest intrinsic rate entrains the other cells of the tissue (Jalife 1984; Verheijck et al. 1998)."

While the cardiac impulse is initiated in the center part of the SAN, gap junctional coupling is lacking there, at least with respect to highly conducting Connexin 43. The presence and any physiological role of other types of connexins in SAN center also remains unclear - see Boyett et al. in *Adv. Cardiol.* 42, 175-197. doi:10.1159/000092569. Furthermore, a new complex mechanism of SAN operation has been recently proposed by Lakatta's group (*JACC: Clin. Electrophysiol.* 2020, 6, 907-931) and Fenske's group (*Nat. Commun.* 2020; 11, 5555). The new paradigm includes interaction of firing and nonfiring cells as well as multi-scale, heterogeneous local calcium signals.

2) Study limitations and data interpretation. You wrote: "Monolayer cultures of cardiac cells beat spontaneously and represent an in vitro model of a natural cardiac pacemaker."

However, contraction elements are scarce or absent in the center of the SA node. Please explain then how your results with blebbistatin can be related to cardiac pacemaker function. Also, as mentioned above, highly conducting Connexin 43 are lacking in the SAN center. SAN cells represent a network of loosely coupled oscillators. Central SAN cells also lack of sodium currents. Thus, your results with culture of strongly coupled ventricular cells may not be applicable for SAN, especially for its central part where the cardiac impulse is initiated. Is it possible then that active force generation plays a role in peripheral area of SAN tissue (or interface to atria)?

3) Methods. About your pharmacological tools:

Please clarify why you specifically chose those blockers vs. others with similar action.

Please discuss non-specific effects of blebbistatin and streptomycin and also ensure that your data interpretation and conclusions are not due to any non-specific effects.

Add a few sentences in introduction or methods about the mechanism of action of blebbistatin and streptomycin.

4) Methods. Data analysis. Your data analysis includes manual inspection and correction of the original data. Please explain how do you avoid (or minimize) possible bias in those procedures. Please provide original recordings with artifacts and after their manual correction.

Also, you filtered the data. Any filtering distorts the data. Please provide examples of your analysis that would clearly show beneficial effects of the filtering.

5) Methods. Data analysis. How activation times were identified from the original recordings? I suggest to make a schematic diagram illustrating all steps and details how you process the original recordings. Also, please provide original recordings.

If you use any computer software, please provide a direct reference/link to the original source of code (GitHub, etc.) so that everyone can reproduce your method.

6) Methods. You wrote: "Due to the limited number of electrodes, it was however not possible to locate the activation focus exactly. Therefore, we resorted to another quantitative marker, which we call slowness, to globally describe the excitation pattern."

Please explain more clearly why you need to introduce the new terms "slowness". Why not to use simply the traditional terms of excitation propagation velocity and/or activation times (i.e. isochrones)?

7) Your cultures. You isolated myocytes and pre-plated them for 2 h to minimize myofibroblast content. However, this procedure still does not exclude cell culture contamination with fibroblasts and other cell types. Is it possible that these contaminations would influence ventricular cell automaticity in your cultures in rest and especially upon stretch and thereby complicate interpretation of your results?

8) Discussion: please formulate more clearly how active force generation can influence the spontaneous activity of your cultures. What is "force-from-filaments principle"?

Would you please give more details of "calcium signalling and nitric oxide" pathway and how this may work in your model? What specific experiments can be performed in the future to find out the detailed mechanism of your blebbistatin effect on spontaneous activity?

Minor issues

1) Introduction. About HRV. You wrote "It may be caused by fluctuations in ion currents due to their stochastic gating (Wilders & Jongsma 1993) or by spatial and temporal variations of ion channel expression and turnover (Ponard et al. 2007)."

It can be also caused by local calcium releases. Please see Monfredi et al. PLoS One. 2013; 8:e67247.

2) Results. To assess complexity, one needs a sufficiently large amount of data. Is your data set large (long) enough to reliably access this parameter? Also, you wrote "Blebbistatin ... significantly reduced the entropy of the distribution of the slowness vector ($p=0.0293$), and thus it decreased the complexity of the variability of the activation pattern."

There is a distinct difference between complexity and entropy. Please clarify

3) Results. Page 11. You wrote: "the slope α of the linear relation between $\log(DF(n))$ and $\log(n)$ was 0.76. This value being greater than 0.5 indicates that the IBI variations exhibit statistically self-similar (fractal) properties."

0.76 is pretty far from near 1 reported for heart rate variability in vivo (as you cited Peng et al. 1995; Behar et al. 2018b). Please explain why you think 0.76 is good enough to make your conclusion.

4) Discussion: It would be nice if you discuss your results with respect to modern concepts of the SAN operation. Please read

Clancy, C. E., and Santana, L. F. (2020). Evolving Discovery of the Origin of the Heartbeat. *JACC: Clin. Electrophysiol.* 6, 932-934. doi:10.1016/j.jacep.2020.07.002

Weiss, J. N., and Qu, Z. (2020). The Sinus Node. *JACC: Clin. Electrophysiol.* 6, 1841-1843. doi:10.1016/j.jacep.2020.09.017

5) Please provide a video of your spontaneously contracting preparation. It would be also nice to see videos of a control culture vs. blebbistatin. Is it possible to see simply by eye the difference in complexity of spontaneous activity, but the same rate (i.e. a major finding of your study)?

6) Please report if similar studies have been performed with other methods, e.g. voltage-sensitive dyes. And if so, was there any difference and why?

7) I think it would be nice if you add a schematic diagram of your results, interpretations, and possible detailed mechanisms.

8) What do you think about a possibility to apply your method directly in freshly isolated or cultured SA node tissue?

9) Please write clear conclusions in the abstract.

10) In methods, please address all question about the use of animals (if appropriate):

"The animals' access to food and water has been described"

"The method of euthanasia has been stated and complies with The Journal's standards"

"Anaesthetic protocols & monitoring have been stated and are appropriate"

"Analgesia and post-operative care in recovery have been stated and are appropriate"

"Terminal procedures have been stated and are appropriate"

Referee #2:

Review of the manuscript entitled "Active force generation contributes to the complexity of spontaneous activity and to the response to stretch of murine cardiomyocyte cultures" by Nayir et al

Using stretchable microelectrode arrays, the authors studied the effects of the myosin inhibitor blebbistatin and the nonselective stretch-activated channel blocker streptomycin on beating variability and on the response of ventricular cell cultures subjected to uniaxial and biaxial deformation. I really enjoyed reading the manuscript which is well written and has very interesting results and interpretation of those results. I was surprised by the lack of effect induced by streptomycin while liked the clarity of the impact of blebbistatin on IBI variability. There interpretation on the role of active force contraction is interesting. It would have been interesting to see how much fibroblast remained in the samples and if that amount was changed because of culturing on PDMS. However, the literature suggests there should not have important changes in density. There is two points that could be addressed in the discussion.

First, although no evidence exists in the literature, there might be conditions for which stretch-activated channels expressions could change. Can "long term" use of streptomycin or other conditions in cell culture influence these expressions? And such changes in expression could impact the lack of response with stretch? It is by itself a study and no requirement for other experiments for this manuscript though. May I suggest the authors verified if anything in the literature could point to this possibility (or point to the impossibility)?

Secondly, it has been suggested that change in effective capacitance could be responsible for changes in conduction velocity (CV) in low stretch amplitude (see PMID 18660447) as referenced by the authors in the introduction. Since CV is changed, conduction delay (slowness) could thus be influenced. It remains to see how to interpret slowness as phase delay in synchronization or conduction delay for a mixture of heterogeneous oscillators and resting excitable cells. I think a discussion on the possible effect of stretch-induced capacitance change should be added.

Minor comments

- Add to fig3. Caption title that this is without deformation (no stretch) for clarity

- Fig 3C and Fig 4: the x label is fuzzy and could be clearer. Actually, all figures seem to have the same problem. It may be because of the journal handling the figures. May I suggest to double check.

- For fig 7B, could it be possible to clarify the y label (or another way than just in the caption), that it is the normalized change in beat rate.

END OF COMMENTS

Confidential Review

10-Mar-2022

Active force generation contributes to the complexity of spontaneous activity and to the response to stretch of murine cardiomyocyte cultures

JP-RP-2022-283083: Revision and Response to Referees

Seyma Nayir, Stéphanie P. Lacour and Jan P. Kucera

We express our gratitude to the editor and the two referees for their positive feedback and for their constructive comments, which assisted us in improving our manuscript. In the following response, our responses are in blue. In our uploaded revision, changes are highlighted using the Track Changes tool of Microsoft Word.

Response regarding required items and journal policies:

Author photo and profile:

We have uploaded a photograph and a short biography of the first author.

Ethical approval and journal guidelines and policies regarding animal experimentation:

We went through the checklists and provide the necessary information, confirmation and citation in our revised manuscript (Methods section, first paragraph “Ethical approval” and paragraph “Murine cardiac cell cultures on sMEAs”).

Statistics policy and Statistical Summary Document:

We went through the checklists to comply with the policy of the Journal. All data points are plotted ($n < 30$). In our revised manuscript, we now mention the number of monolayer preparations as well as the number of litters from which the cardiomyocytes were obtained. All “n” values are clearly stated in the text or the figure legends. Moreover, p values with 3 significant digits are now also provided even in the absence of statistical significance. The statistics summary Excel table was completed and uploaded.

Abstract Figure:

We have prepared an Abstract Figure using BioRender and the corresponding legend.

Response to the Reviewing Editor:

Both reviewers found the manuscript interesting but both highlighted issues that need addressing before potential reconsideration. The importance of these results for normal physiology and/or pathophysiology also needs clarifying in the revised Discussion.

We express our gratitude to the Reviewing Editor for appreciating of our work. In our revised manuscript, we positioned our work in the context of the literature suggested by Referee #1 (for which we are very thankful), and we believe that this has clarified the meaning of our results for physiology and pathophysiology.

Response to the Senior Editor:

I concur with the reviewing editor. In your revision, please also consider that blebbistatin will reduce ATP consumption and hence the metabolic load on the cells. This effect should be discussed.

We thank the Senior Editor for this feedback. As mentioned in our response to the Reviewing Editor, we believe that positioning our work in the context of the literature suggested by Referee #1 has clarified the meaning of our results for physiology and pathophysiology.

In our revised discussion (section Limitations), we discuss the effect of blebbistatin on ATP consumption and metabolic load in the context of non-specific effects of blebbistatin. Indeed, increased ATP availability may potentiate the function of sarcolemmal and sarcoplasmic pumps, and decreased metabolic load may modify the function of mitochondria, which are organelles that participate in cardiomyocyte Ca^{2+} homeostasis (Shannon & Bers 2004). These considerations about ATP consumption and metabolism will apply to any contraction uncoupler.

Response to Referee #1:

Using stretchable microelectrode arrays, Nayir et al. investigates complexity of spontaneous activity in murine cardiomyocyte cultures with respect to stretch. The authors apply pharmacological tools to explore two mechanisms that could modulate the activity: active force generation and stretch. The paper is interesting, original, and technically well done. However, the importance of these results for physiology remains unclear to me. I also have several questions about their methods, robustness of the data, and data interpretation.

We express our gratitude to Referee #1 for his/her appreciation of our work. We believe that by elaborating on our work in the context of the literature suggested by the Referee (for which we are thankful) has rendered the meaning of our results for physiology clearer.

Major issues.

1) Introduction. You wrote: Pacemaker cells synchronize via gap junctional coupling by mutual entrainment, whereby the cells or a group of cells exhibiting the fastest intrinsic rate entrains the other cells of the tissue (Jalife 1984; Verheijck et al. 1998)."

While the cardiac impulse is initiated in the center part of the SAN, gap junctional coupling is lacking there, at least with respect to highly conducting Connexin 43. The presence and any physiological role of other types of connexins in SAN center also remains unclear - see Boyett et al. in *Adv. Cardiol.* 42, 175-197. doi:10.1159/000092569 (Boyett *et al.* 2000). Furthermore, a new complex mechanism of SAN operation has been recently proposed by Lakatta's group (*JACC: Clin. Electrophysiol.* 2020, 6, 907-931) (Bychkov *et al.* 2020) and Fenske's group (*Nat. Commun.* 2020; 11, 5555) (Fenske *et al.* 2020). The new paradigm includes interaction of firing and nonfiring cells as well as multi-scale, heterogeneous local calcium signals.

We thank the referee for rendering us attentive to this very important work. In our revised introduction, we present extensively the salient findings of these studies and introduce our work in the context of this new paradigm.

2) Study limitations and data interpretation. You wrote: "Monolayer cultures of cardiac cells beat spontaneously and represent an in vitro model of a natural cardiac pacemaker."

We have toned down this statement in the abstract and the key points summary by writing "Cardiomyocyte cultures / monolayer cultures of cardiac cells exhibit spontaneous electrical and contractile activity, as in a natural cardiac pacemaker".

However, contraction elements are scarce or absent in the center of the SA node. Please explain then how your results with blebbistatin can be related to cardiac pacemaker function. Also, as mentioned

above, highly conducting Connexin 43 are lacking in the SAN center. SAN cells represent a network of loosely coupled oscillators. Central SAN cells also lack of sodium currents. Thus, your results with culture of strongly coupled ventricular cells may not be applicable for SAN, especially for its central part where the cardiac impulse is initiated. Is it possible then that active force generation plays a role in peripheral area of SAN tissue (or interface to atria)?

As shown in the study of Bychkov et al. (Bychkov *et al.* 2020), the SAN consists of intertwined F-actin positive and negative tissue. Thus, contractile elements are not completely absent in the centre of the SA node. Moreover, both active contractile and passive elastic forces can be transmitted over a large distance (several cells) due to the action-reaction principle (Newton's 3rd law). We agree with the reviewer that the role of active force generation may be more important in the periphery of the SA node. Regarding the paucity of connexin 43 and sodium channels in the SAN centre, we agree that our *in vitro* model suffers from this limitation. We have integrated these considerations in our revised discussion (first paragraph of the "Limitations" section).

3) Methods. About your pharmacological tools:

Please clarify why you specifically chose those blockers vs. others with similar action.

We selected blebbistatin because this agent is widely used by the cardiac optical mapping community to suppress motion artefacts (e.g., (Lou *et al.* 2012; Quinn *et al.* 2017), including (Fenske *et al.* 2020)) and exerts less effects on electrophysiological parameters compared to other agents such as 2,3-butanedione monoxime, diacetyl monoxime or cytochalasin D which are known to modify the cardiac action potential (Baker *et al.* 2004; Brines *et al.* 2012; Lou *et al.* 2012).

We selected streptomycin because it is a common pharmacological tool to investigate stretch-activated channels in cardiac electrophysiology (White 2006; Quinn *et al.* 2017; Quinn & Kohl 2021) and it has been successfully used in cardiac cell culture systems similar to ours (Thompson *et al.* 2011; Grand *et al.* 2014). Other broadly used agents are gadolinium and GsMTx-4. However, gadolinium precipitates in the presence of phosphate and bicarbonate (Quinn & Kohl 2021), precluding its use in the medium that we used for experiments (HBSS). It may also block the sodium current (Quinn & Kohl 2021). As we observed no effects with streptomycin, it was not in our scope to additionally test GsMTx-4.

We have incorporated these aspects in the Methods section.

Please discuss non-specific effects of blebbistatin and streptomycin and also ensure that your data interpretation and conclusions are not due to any non-specific effects.

In our study, we used blebbistatin to block active force generation. Nonetheless, it cannot be excluded that blebbistatin exerts non-specific effects, either directly on ion channels or Ca²⁺ handling proteins, or indirectly by decreasing ATP consumption and the cellular metabolic load. Increased ATP availability may potentiate the function of sarcolemmal and sarcoplasmic pumps, and inhibit the ATP-sensitive potassium current (Swift *et al.* 2021). Decreased metabolic load may modify the function of mitochondria, which are organelles that participate in cardiomyocyte Ca²⁺ homeostasis (Shannon & Bers 2004). These considerations about metabolism will apply to any contraction uncoupler.

In our study, we used streptomycin, which may also block potassium channels and L-type Ca²⁺ channels (Quinn & Kohl 2021). Based on our previous observation that blocking L-type Ca²⁺ channels with nifedipine completely suppresses spontaneous electrical activity in cardiomyocyte cultures (Ponard *et al.* 2007), such a non-specific effect would likely have depressed beat rate in our preparations, which we did not observe.

We incorporated these aspects in the Limitations section of the discussion.

Add a few sentences in introduction or methods about the mechanism of action of blebbistatin and streptomycin.

Blebbistatin is a myosin II-specific inhibitor that blocks the cross-bridge cycle by binding to an allosteric site in the motor domain of myosin II. It stabilizes an actin-detached state of myosin II, which prevents it from generating force and using ATP (Rauscher *et al.* 2018; Roman *et al.* 2018).

Streptomycin is an aminoglycoside antibiotic that also blocks non-specific stretch-activated channels (Quinn & Kohl 2021).

We have incorporated this in the Methods section.

4) Methods. Data analysis. Your data analysis includes manual inspection and correction of the original data. Please explain how do you avoid (or minimize) possible bias in those procedures. Please provide original recordings with artifacts and after their manual correction.

In the revised manuscript (Methods section), we now provide more details regarding the analysis and illustrate it in Figure 1B. The most important aspect in minimizing bias was to use the lowest possible threshold for detection to ensure a high sensitivity (low number of false negative detections that needed to be corrected manually) at the expense of a lower specificity (larger number of detections that needed to be deleted manually, but which does not introduce bias).

We note that consciously (deliberately) or unconsciously altering (biasing) the activation times in order to influence a main result of our study (e.g., the results in Figure 3C) would require a priori knowledge of how a given alteration impacts on the corresponding result (e.g., entropy or detrended fluctuation analysis exponent). Because we did not have such a priori knowledge, this kind of bias is absent. Moreover, this would require alteration of tens of thousands of activation times, which is neither realistic nor feasible.

Also, you filtered the data. Any filtering distorts the data. Please provide examples of your analysis that would clearly show beneficial effects of the filtering.

Signals had a typical biphasic shape (see examples in Figures 1B, 1C and 1D), with an initial positive phase, a rapid downstroke lasting typically <1 ms and a negative phase. Because this time course occurs on a time scale shorter than the time constant of the AC filter and because the Kaiser filter (used for low-pass filtering) involved a symmetric kernel (producing, by design, no phase shifts in the frequency domain), the resulting effect on activation time detection was minimal. We clarified this in the revised text. In Figure 1B, we show examples of filtering.

5) Methods. Data analysis. How activation times were identified from the original recordings? I suggest to make a schematic diagram illustrating all steps and details how you process the original recordings. Also, please provide original recordings.

As stated above, we now provide more details regarding the analysis in the Methods section and illustrate it schematically in Figure 1B. The original data are available on the repository Zenodo (see below).

Of note, in the online supplement to Ponard *et al.* (2007, *op. cit.*), we assessed in computer simulations the effect of noise on the temporal accuracy of activation time determination in extracellular recordings from monolayer cardiomyocyte cultures. The simulations showed that the activation time could be detected with an accuracy of 0.1 ms, which increases to 0.3 ms only for very noisy signals. To ascertain the robustness of our present results, we conducted a simulation in which all the activation times were corrupted by a random value drawn uniformly from the interval [-1 ms, 1 ms], which is already a substantial error. We then performed the analyses shown in Figures 2 and 3. The results were essentially similar, with the most notable difference that the 2 principal components could not explain >95% of the variance (a fact that can be straightforwardly explained by the introduction of noise).

However, and importantly, the significance of the results shown in Figure 3C regarding entropy and Spearman's rank correlation were not affected.

If you use any computer software, please provide a direct reference/link to the original source of code (GitHub, etc.) so that everyone can reproduce your method.

All raw recordings as well as the MATLAB source code of the user interface and scripts generating the panels of Figures 2-8 were deposited on the repository Zenodo and are openly available (<https://doi.org/10.5281/zenodo.6524580>). This is now stated at the end of the Methods section and in the Data Availability Statement.

6) Methods. You wrote: "Due to the limited number of electrodes, it was however not possible to locate the activation focus exactly. Therefore, we resorted to another quantitative marker, which we call slowness, to globally describe the excitation pattern."

Please explain more clearly why you need to introduce the new terms "slowness". Why not to use simply the traditional terms of excitation propagation velocity and/or activation times (i.e. isochrones)?

In activation maps (irrespective of whether they are obtained using optical mapping or with an extracellular electrode array), to evaluate conduction velocity, one typically fits (regresses) a set of activation times vs. positions. In this setting, the independent variable (the one that is given, or controlled by the investigator) is position, whereas the dependent variable (the one that is measured by the investigator) is time. In regression analyses, one regresses the dependent vs. the independent variable (see e.g. https://en.wikipedia.org/wiki/Regression_analysis and the references therein). As *primary* output from the regression, one obtains the slope of the fitted line (or curve, or surface), which, in our case is slowness and has units of time per distance. The traditional excitation propagation velocity (conduction velocity) is then obtained as a *secondary* output by taking the inverse of slowness. Thus, in a first step, slowness is computed as the spatial gradient of activation time, and conduction velocity is computed from slowness in a subsequent step. This is exactly the approach used by others (Masè *et al.* 2021; van Schie *et al.* 2021), and, importantly, in the widely used algorithm of Bayly *et al.* (Bayly *et al.* 1998). Note that this differs from an experiment in which, for example, the position of an object is followed using stroboscopic light pulses. In this setting, time is the independent variable, position is the dependent variable, and the result of the regression is indeed velocity. As another example, consider a patch clammer recording the current through an ion channel in the voltage clamp mode. The independent variable is the clamped voltage, the dependent variable is the measured current, the outcome of the regression procedure is conductance, and the result is presented as conductance, not resistance.

These considerations justify the use of slowness because it is the primary output of the regression procedure. A second argument in favour of slowness is that if differences in activation times are small, or, if the excitation pattern is symmetric (e.g., a radially symmetric pattern originating in the centre of the mapped region), overall slowness will be very small (or even 0) and velocity will diverge towards infinity (division by 0). The use of slowness permits to avoid this singularity.

We have clarified the use of slowness in the Methods section.

Our user interface uploaded on Zenodo shows both slowness and conduction velocity.

Isochrones are graphical objects that help investigators to visualize the activation sequence in a map, but, by themselves, they do not provide any quantitative information. The concept of "activation times" is used in our work in the traditional sense.

7) Your cultures. You isolated myocytes and pre-plated them for 2 h to minimize myofibroblast content. However, this procedure still does not exclude cell culture contamination with fibroblasts and other

cell types. Is it possible that these contaminations would influence ventricular cell automaticity in your cultures in rest and especially upon stretch and thereby complicate interpretation of your results?

The Referee indeed addresses a very important point, and in our initial manuscript, we had already discussed this in a dedicated paragraph of the discussion entitled "Is it possible that myofibroblasts caused the observed effects?".

In the revision of this paragraph, we now mention the other cell types.

Of note, cardiac myofibroblasts in culture systems require the presence of serum to survive and experiments with such cells are typically conducted with 1% of neonatal calf serum (Miragoli *et al.* 2007; Thompson *et al.* 2011; Grand *et al.* 2014). Our recordings were conducted in the absence of serum, which further minimizes the possibility that myofibroblasts may have affected our observations. We have added this note in that same paragraph.

8) Discussion: please formulate more clearly how active force generation can influence the spontaneous activity of your cultures. What is "force-from-filaments principle"? Would you please give more details of "calcium signalling and nitric oxide" pathway and how this may work in your model? What specific experiments can be performed in the future to find out the detailed mechanism of your blebbistatin effect on spontaneous activity?

As reviewed by Boycott *et al.* (Boycott *et al.* 2020), nitric oxide (NO), produced in cardiomyocytes by both endothelial and neuronal NO synthases, is influenced by mechanical forces. In turn, NO impacts on the function of integrins, which link the actin cytoskeleton to the extracellular matrix. NO also influences further cytoskeletal proteins (e.g., talin, vinculin) that are able to sense and integrate mechanical signals. It was suggested that NO stimulation of integrins then promotes NO-mediated Ca^{2+} release from the sarcoplasmic reticulum and modulate the function of L-type Ca^{2+} channels (Boycott *et al.* 2020). In the context of pacemaker function, this would exert repercussion on the Ca^{2+} clock and affect beat rate. To investigate the involvement of NO in mediating the effects of active force on the response to stretch, experiments could be performed in the future in which NO production is depressed by the NO synthase inhibitor L-NAME or enhanced by the NO donor SNAP (Boycott *et al.* 2020) in the presence vs. absence of blebbistatin.

We have incorporated this text into the discussion.

The "force from lipids" and "force from filaments" principles are notions that were popularized by Martinac *et al.* (e.g., (Cox *et al.* 2019)) in which ion channels sense stretch directly via membrane phospholipids or indirectly via cytoskeletal proteins or extracellular tethers with which the channels interact. In the context of our findings, it appears more plausible (if ion channels are involved), that the influence of active force generation is mediated by the force-from-filaments principle.

We have clarified this in our discussion.

Minor issues

1) Introduction. About HRV. You wrote "It may be caused by fluctuations in ion currents due to their stochastic gating (Wilders & Jongsma 1993) or by spatial and temporal variations of ion channel expression and turnover (Ponard *et al.* 2007)."

It can be also caused by local calcium releases. Please see Monfredi *et al.* PLoS One. 2013; 8:e67247.

We fully agree with the Referee and we revised this sentence accordingly.

2) Results. To assess complexity, one needs a sufficiently large amount of data. Is your data set large (long) enough to reliably access this parameter? Also, you wrote "Blebbistatin ... significantly reduced the entropy of the distribution of the slowness vector ($p=0.0293$), and thus it decreased the complexity of the variability of the activation pattern."

There is a distinct difference between complexity and entropy. Please clarify

Although there exist distinct mathematical methods to quantify complexity (e.g., (Lempel & Ziv 1976)), we use the words “complex” and “complexity” in their literal and intuitive meaning as “complicated, not easy to describe or to understand”, as opposed to “simple”. We clarified this in the Methods section.

The experiments presented in Figure 3 involved 547 ± 246 activations per recording and are in our opinion sufficiently long to quantify entropy as proposed.

3) Results. Page 11. You wrote: "the slope α of the linear relation between $\log(\text{DF}(n))$ and $\log(n)$ was 0.76. This value being greater than 0.5 indicates that the IBI variations exhibit statistically self-similar (fractal) properties."

0.76 is pretty far from near 1 reported for heart rate variability in vivo (as you cited Peng et al. 1995; Behar et al. 2018b). Please explain why you think 0.76 is good enough to make your conclusion.

What is important here is that it is >0.5 , which is what would be expected for white noise. This allows inferring that the IBI variations exhibit statistically self-similar (fractal) properties distinct from random noise. The fact that it is <1 suggests that in vivo, further factors are likely to be involved in shaping heart rate variability. We have clarified this in the results section and at the beginning of the discussion.

4) Discussion: It would be nice if you discuss your results with respect to modern concepts of the SAN operation. Please read

Clancy, C. E., and Santana, L. F. (2020). Evolving Discovery of the Origin of the Heartbeat. *JACC: Clin. Electrophysiol.* 6, 932-934. doi:10.1016/j.jacep.2020.07.002

Weiss, J. N., and Qu, Z. (2020). The Sinus Node. *JACC: Clin. Electrophysiol.* 6, 1841-1843. doi:10.1016/j.jacep.2020.09.017

We thank the Referee for this suggestion and we have read these articles with great interest. They go hand in hand with the literature mentioned under the Referee’s major issue 1. For this reason, we thought it would be more appropriate to discuss them in our introduction, where we present the most recent concepts of SAN operation. In addition, we conclude our discussion as follows:

“Recent research has challenged the view that a dominant centre in the SAN drives pacemaking (Clancy & Santana 2020), suggested that the SAN operates as a critical system (Weiss & Qu 2020) and raised the question whether factors others than gap junctional coupling are involved in impulse formation in the SAN (Weiss & Qu 2020). Thus, our results suggest that mechanical mechanisms represent a path worth to explore.”

5) Please provide a video of your spontaneously contracting preparation. It would be also nice to see videos of a control culture vs. blebbistatin. Is it possible to see simply by eye the difference in complexity of spontaneous activity, but the same rate (i.e. a major finding of your study)?

We provide such a video showing a contracting monolayer under control conditions, upon application of blebbistatin and after washout. The video shows that blebbistatin fully blocks the contractions of the myocytes but that the effect of blebbistatin cannot be washed out. Beat rate variability can be appreciated by eye under control conditions, although the spread of the excitation wave is too fast to be seen. However, it is not possible to see by eye the difference in complexity of spontaneous activity with blebbistatin because the preparation does not contract any more.

6) Please report if similar studies have been performed with other methods, e.g. voltage-sensitive dyes. And if so, was there any difference and why?

To our knowledge, similar experiments have not been performed with voltage-sensitive dyes. Based on the corresponding's author past experience with the rapidly reacting voltage-sensitive dye di-8-ANEPPS in cardiomyocyte cultures (Rohr & Kucera 1998; Rohr *et al.* 1998), such dyes cause phototoxic damage to the preparations within seconds to a couple of minutes, precluding long-term experiments like in the present study. Optogenetic reporters of membrane potential are probably still too slow at the present stage to offer sufficient temporal resolution.

Experiments (lasting up to 30 s) using cardiomyocyte monolayers grown on PDMS have been conducted by Boudreau-Béland *et al.* (Boudreau-Béland *et al.* 2015) and the similarities and differences were already discussed in our initial manuscript.

To our knowledge, calcium mapping experiments at the cellular or subcellular scale (such as those of Bychkov *et al.* (Bychkov *et al.* 2020) and Fenske *et al.* (Fenske *et al.* 2020)) have not been reported for cardiomyocyte monolayers.

7) I think it would be nice if you add a schematic diagram of your results, interpretations, and possible detailed mechanisms.

We have prepared such a schematic as a Graphical Abstract.

8) What do you think about a possibility to apply your method directly in freshly isolated or cultured SA node tissue?

Such a possibility would indeed be nice, as it would certainly provide further insights into real SA node function. Ideally, SA nodes of larger mammals should be used. However, we did not have access to such tissues. The challenges that we foresee will be to ensure that the contact between the tissue and the sMEA remains tight and that the tissue closely follows the movement of the sMEA upon stretch. This challenge will require further developments in bioengineering.

We have incorporated these considerations in the Limitations section.

9) Please write clear conclusions in the abstract.

We have deleted the last sentence of the abstract, which may be perceived as too vague and speculative, and clarified the concluding statement of the abstract, which now reads as follows:

"Therefore, our data support the notion that in a spontaneously firing network of cardiomyocytes, active force generation, rather than stretch-activated channels, is mechanistically involved in the complexity of the spatiotemporal patterns of spontaneous activity and in the stretch-induced acceleration of beating."

10) In methods, please address all question about the use of animals (if appropriate):

"The animals' access to food and water has been described"

"The method of euthanasia has been stated and complies with The Journal's standards"

"Anaesthetic protocols & monitoring have been stated and are appropriate"

"Analgesia and post-operative care in recovery have been stated and are appropriate"

"Terminal procedures have been stated and are appropriate"

We went through this checklist and as per Journal policy, we provided the necessary information, confirmation and citation of the article by David Grundy in our revised manuscript (Methods section, first paragraph "Ethical approval" and paragraph "Murine cardiac cell cultures on sMEAs").

Response to Referee #2:

Review of the manuscript entitled "Active force generation contributes to the complexity of spontaneous activity and to the response to stretch of murine cardiomyocyte cultures" by Nayir et al

Using stretchable microelectrode arrays, the authors studied the effects of the myosin inhibitor blebbistatin and the nonselective stretch-activated channel blocker streptomycin on beating variability and on the response of ventricular cell cultures subjected to uniaxial and biaxial deformation. I really enjoyed reading the manuscript which is well written and has very interesting results and interpretation of those results. I was surprised by the lack of effect induced by streptomycin while liked the clarity of the impact of blebbistatin on IBI variability. The interpretation on the role of active force contraction is interesting. It would have been interesting to see how much fibroblast remained in the samples and if that amount was changed because of culturing on PDMS. However, the literature suggests there should not have important changes in density. There is two points that could be addressed in the discussion.

We express our gratitude to Referee #2 for his/her positive feedback on our work. It would indeed have been interesting to see how many residual fibroblasts remained in the cultures. We did not have access to immunohistological assays.

Of note, cardiac myofibroblasts in culture systems require the presence of serum to survive and experiments with such cells are typically conducted with 1% of neonatal calf serum (Miragoli *et al.* 2007; Thompson *et al.* 2011; Grand *et al.* 2014). Our recordings were conducted in the absence of serum, which further minimizes the possibility that myofibroblasts may have affected our observations. We have added this comment in our revised discussion.

First, although no evidence exists in the literature, there might be conditions for which stretch-activated channels expressions could change. Can "long term" use of streptomycin or other conditions in cell culture influence these expressions? And such changes in expression could impact the lack of response with stretch? It is by itself a study and no requirement for other experiments for this manuscript though. May I suggest the authors verified if anything in the literature could point to this possibility (or point to the impossibility)?

Our search in PubMed using the keywords "streptomycin" "stretch" "channels" and "expression" returned no pertinent result regarding the expression of stretch-activated channels under long-term exposure to streptomycin. In the revised discussion, we elaborate on the hypothesis raised by the Referee as follows (3rd paragraph of the Limitations section):

"There may also be conditions for which the expression of stretch-activated channels could change, e.g., by streptomycin exposure during the culture period preceding the experiments, which may contribute to the absence of streptomycin effects. However, we found no specific evidence in the literature that streptomycin up- or downregulates the expression of SACs."

Secondly, it has been suggested that change in effective capacitance could be responsible for changes in conduction velocity (CV) in low stretch amplitude (see PMID 18660447) (Mills *et al.* 2008) as referenced by the authors in the introduction. Since CV is changed, conduction delay (slowness) could thus be influenced. It remains to see how to interpret slowness as phase delay in synchronization or conduction delay for a mixture of heterogeneous oscillators and resting excitable cells. I think a discussion on the possible effect of stretch-induced capacitance change should be added.

Based on experiments in volume loaded ventricles and computational analyses, Mills *et al.* (Mills *et al.* 2008) indeed suggested that stretch could lead to an increase in membrane capacitance. The subsequent work of Pfeiffer *et al.* (Pfeiffer *et al.* 2014) indicates that stretch increases the capacitance of cultured ventricular myocytes via recruitment of caveolae to the sarcolemma. Assuming that the

number and function of membrane channels and transporters is not affected by stretch, the same ion currents would then have to charge/discharge a larger capacitive load, which is expected to slow both depolarization and repolarization, and thus spontaneous beat rate. However, we did not observe this in our study. As the time course of caveolae incorporation and possible capacitance changes is unknown, we can nevertheless speculate that an increase in capacitance could have contributed to the slow return of beat rate towards baseline after a few seconds (see Figures 5-8). Further studies with dedicated systems, such as the platform developed by De Coulon et al. (de Coulon *et al.* 2021) may provide an appropriate answer in the future.

We have integrated these considerations in the revised discussion (3rd paragraph).

Regarding the interpretation of slowness as phase delay in synchronization or conduction delay for a mixture of heterogeneous oscillators and resting excitable cells: Such an interpretation is not feasible because we considered slowness as a macroscopic scale marker and our stretchable MEAs do not permit to discriminate between firing from resting cells.

Minor comments

- Add to fig3. Caption title that this is without deformation (no stretch) for clarity

We thank the Referee for this suggestion- We have added “without external deformation” to the figure title.

- Fig 3C and Fig 4: the x label is fuzzy and could be clearer. Actually, all figures seem to have the same problem. It may be because of the journal handling the figures. May I suggest to double check.

We increased the font size of the x labels in these two Figures as well as in Figures 7B and 8B. It is possible that this problem has arisen from the compression of the figures embedded in the manuscript PDF file generated by the server of the Journal. We uploaded separately high-resolution (1200 dpi) TIFF files, which we double-checked, and which should be available to the Referees. In any case, we will make sure with the publisher that our figures are rendered correctly.

As an alternative, we tested vectorised image files, but the problem is that usual PDF viewers are much too slow to render some of the figures, which consist of tens of thousands of points and line segments.

- For fig 7B, could it be possible to clarify the y label (or another way than just in the caption), that it is the normalized change in beat rate.

We have renamed the corresponding y labels in Figures 7B and 8B as “Fold Change in Beat Rate” to make this clearer, also in the legends. Moreover, we removed the unit “Hz” for normalized beat rate in Figures 5B, 5C, 6A, 7A and 8A, which was clearly erroneous (and confusing) because this quantity is unitless after normalization. We thank the Referee for this comment, as it permitted us to identify our mistake.

References

Baker LC, Wolk R, Choi BR, Watkins S, Plan P, Shah A & Salama G (2004). Effects of mechanical uncouplers, diacetyl monoxime, and cytochalasin-D on the electrophysiology of perfused mouse hearts. *Am J Physiol Heart Circ Physiol* **287**, H1771-1779.

Bayly PV, KenKnight BH, Rogers JM, Hillsley RE, Ideker RE & Smith WM (1998). Estimation of conduction velocity vector fields from epicardial mapping data. *IEEE Trans Biomed Eng* **45**, 563-571.

- Boudreau-Béland J, Duverger JE, Petitjean E, Maguy A, Ledoux J & Comtois P (2015). Spatiotemporal stability of neonatal rat cardiomyocyte monolayers spontaneous activity is dependent on the culture substrate. *PLoS One* **10**, e0127977.
- Boycott HE, Nguyen MN, Vrellaku B, Gehmlich K & Robinson P (2020). Nitric oxide and mechano-electrical transduction in cardiomyocytes. *Front Physiol* **11**, 606740.
- Boyett MR, Honjo H & Kodama I (2000). The sinoatrial node, a heterogeneous pacemaker structure. *Cardiovasc Res* **47**, 658-687.
- Brines L, Such-Miquel L, Gallego D, Trapero I, Del Canto I, Zarzoso M, Soler C, Pelechano F, Canoves J, Alberola A, Such L & Chorro FJ (2012). Modifications of mechanoelectric feedback induced by 2,3-butanedione monoxime and blebbistatin in Langendorff-perfused rabbit hearts. *Acta Physiol (Oxf)* **206**, 29-41.
- Bychkov R, Juhaszova M, Tsutsui K, Coletta C, Stern MD, Maltsev VA & Lakatta EG (2020). Synchronized cardiac impulses emerge from heterogeneous local calcium signals within and among cells of pacemaker tissue. *JACC Clin Electrophysiol* **6**, 907-931.
- Clancy CE & Santana LF (2020). Evolving discovery of the origin of the heartbeat: a new perspective on sinus rhythm. *JACC Clin Electrophysiol* **6**, 932-934.
- Cox CD, Bavi N & Martinac B (2019). Biophysical principles of ion-channel-mediated mechanosensory transduction. *Cell Rep* **29**, 1-12.
- de Coulon E, Dellenbach C & Rohr S (2021). Advancing mechanobiology by performing whole-cell patch clamp recording on mechanosensitive cells subjected simultaneously to dynamic stretch events. *iScience* **24**, 102041.
- Fenske S, Hennis K, Rotzer RD, Brox VF, Becirovic E, Scharr A, Gruner C, Ziegler T, Mehlfeld V, Brennan J, Efimov IR, Pauza AG, Moser M, Wotjak CT, Kupatt C, Gonner R, Zhang R, Zhang H, Zong X, Biel M & Wahl-Schott C (2020). cAMP-dependent regulation of HCN4 controls the tonic entrainment process in sinoatrial node pacemaker cells. *Nat Commun* **11**, 5555.
- Grand T, Salvarani N, Jousset F & Rohr S (2014). Aggravation of cardiac myofibroblast arrhythmogenicity by mechanical stress. *Cardiovasc Res* **104**, 489-500.
- Lempel A & Ziv J (1976). On the complexity of finite sequences. *IEEE Trans Inf Theory* **22**, 75-81.
- Lou Q, Li W & Efimov IR (2012). The role of dynamic instability and wavelength in arrhythmia maintenance as revealed by panoramic imaging with blebbistatin vs. 2,3-butanedione monoxime. *Am J Physiol Heart Circ Physiol* **302**, H262-269.
- Masè M, Cristoforetti A, Del Greco M & Ravelli F (2021). A divergence-based approach for the identification of atrial fibrillation focal drivers from multipolar mapping: a computational study. *Front Physiol* **12**, 749430.
- Mills RW, Narayan SM & McCulloch AD (2008). Mechanisms of conduction slowing during myocardial stretch by ventricular volume loading in the rabbit. *Am J Physiol Heart Circ Physiol* **295**, H1270-H1278.

- Miragoli M, Salvarani N & Rohr S (2007). Myofibroblasts induce ectopic activity in cardiac tissue. *Circ Res* **101**, 755-758.
- Pfeiffer ER, Wright AT, Edwards AG, Stowe JC, McNall K, Tan J, Niesman I, Patel HH, Roth DM, Omens JH & McCulloch AD (2014). Caveolae in ventricular myocytes are required for stretch-dependent conduction slowing. *J Mol Cell Cardiol* **76**, 265-274.
- Ponard JG, Kondratyev AA & Kucera JP (2007). Mechanisms of intrinsic beating variability in cardiac cell cultures and model pacemaker networks. *Biophys J* **92**, 3734-3752.
- Quinn TA, Jin H, Lee P & Kohl P (2017). Mechanically induced ectopy via stretch-activated cation-nonspecific channels is caused by local tissue deformation and results in ventricular fibrillation if triggered on the repolarization wave edge (commotio cordis). *Circ Arrhythm Electrophysiol* **10**.
- Quinn TA & Kohl P (2021). Cardiac mechano-electric coupling: acute effects of mechanical stimulation on heart rate and rhythm. *Physiol Rev* **101**, 37-92.
- Rauscher AA, Gyimesi M, Kovács M & Málnási-Csizmadia A (2018). Targeting myosin by blebbistatin derivatives: optimization and pharmacological potential. *Trends Biochem Sci* **43**, 700-713.
- Rohr S & Kucera JP (1998). Optical recording system based on a fiberoptic image conduit: assessment of microscopic activation patterns in cardiac tissue. *Biophys J* **75**, 1062-1075.
- Rohr S, Kucera JP & Kléber AG (1998). Slow conduction in cardiac tissue, I: effects of a reduction of excitability versus a reduction of electrical coupling on microconduction. *Circ Res* **83**, 781-794.
- Roman BI, Verhasselt S & Stevens CV (2018). Medicinal chemistry and use of myosin II inhibitor (S)-blebbistatin and its derivatives. *J Med Chem* **61**, 9410-9428.
- Shannon TR & Bers DM (2004). Integrated Ca²⁺ management in cardiac myocytes. *Ann N Y Acad Sci* **1015**, 28-38.
- Swift LM, Kay MW, Ripplinger CM & Posnack NG (2021). Stop the beat to see the rhythm: excitation-contraction uncoupling in cardiac research. *Am J Physiol Heart Circ Physiol* **321**, H1005-H1013.
- Thompson SA, Copeland CR, Reich DH & Tung L (2011). Mechanical coupling between myofibroblasts and cardiomyocytes slows electric conduction in fibrotic cell monolayers. *Circulation* **123**, 2083-2093.
- van Schie MS, Heida A, Taverne Y, Bogers A & de Groot NMS (2021). Identification of local atrial conduction heterogeneities using high-density conduction velocity estimation. *Europace* **23**, 1815-1825.
- Weiss JN & Qu Z (2020). The sinus node: still mysterious after all these years. *JACC Clin Electrophysiol* **6**, 1841-1843.
- White E (2006). Mechanosensitive channels: therapeutic targets in the myocardium? *Curr Pharm Des* **12**, 3645-3663.

Dear Dr Kucera,

Re: JP-RP-2022-283083R1 "Active force generation contributes to the complexity of spontaneous activity and to the response to stretch of murine cardiomyocyte cultures" by Seyma Nayir, Stéphanie P Lacour, and Jan P Kucera

I am pleased to tell you that your paper has been accepted for publication in The Journal of Physiology.

NEW POLICY: In order to improve the transparency of its peer review process The Journal of Physiology publishes online as supporting information the peer review history of all articles accepted for publication. Readers will have access to decision letters, including all Editors' comments and referee reports, for each version of the manuscript and any author responses to peer review comments. Referees can decide whether or not they wish to be named on the peer review history document.

The last Word version of the paper submitted will be used by the Production Editors to prepare your proof. When this is ready you will receive an email containing a link to Wiley's Online Proofing System. The proof should be checked and corrected as quickly as possible.

Authors should note that it is too late at this point to offer corrections prior to proofing. The accepted version will be published online, ahead of the copy edited and typeset version being made available. Major corrections at proof stage, such as changes to figures, will be referred to the Reviewing Editor for approval before they can be incorporated. Only minor changes, such as to style and consistency, should be made a proof stage. Changes that need to be made after proof stage will usually require a formal correction notice.

All queries at proof stage should be sent to TJP@wiley.com

Are you on Twitter? Once your paper is online, why not share your achievement with your followers. Please tag The Journal (@jphysiol) in any tweets and we will share your accepted paper with our 23,000+ followers!

Yours sincerely,

Bjorn Knollmann
Senior Editor
The Journal of Physiology

P.S. - You can help your research get the attention it deserves! Check out Wiley's free Promotion Guide for best-practice recommendations for promoting your work at www.wileyauthors.com/eeo/guide. And learn more about Wiley Editing Services which offers professional video, design, and writing services to create shareable video abstracts, infographics, conference posters, lay summaries, and research news stories for your research at www.wileyauthors.com/eeo/promotion.

*** IMPORTANT NOTICE ABOUT OPEN ACCESS ***

To assist authors whose funding agencies mandate public access to published research findings sooner than 12 months after publication The Journal of Physiology allows authors to pay an open access (OA) fee to have their papers made freely available immediately on publication.

You will receive an email from Wiley with details on how to register or log-in to Wiley Authors Services where you will be able to place an OnlineOpen order.

You can check if you funder or institution has a Wiley Open Access Account here <https://authorservices.wiley.com/author-resources/Journal-Authors/licensing-and-open-access/open-access/author-compliance-tool.html>

Your article will be made Open Access upon publication, or as soon as payment is received.

If you wish to put your paper on an OA website such as PMC or UKPMC or your institutional repository within 12 months of publication you must pay the open access fee, which covers the cost of publication.

OnlineOpen articles are deposited in PubMed Central (PMC) and PMC mirror sites. Authors of OnlineOpen articles are permitted to post the final, published PDF of their article on a website, institutional repository, or other free public server, immediately on publication.

Note to NIH-funded authors: The Journal of Physiology is published on PMC 12 months after publication, NIH-funded authors DO NOT NEED to pay to publish and DO NOT NEED to post their accepted papers on PMC.

EDITOR COMMENTS

Reviewing Editor:

No further comments.

Senior Editor:

I concur with the reviewing editor. Excellent work!

REFEREE COMMENTS

Referee #1:

The authors have addressed all my original comments in their revised paper version. The paper has been substantially improved.

Referee #2:

The modifications made by the authors are to my satisfaction.

1st Confidential Review

17-May-2022